# PERFORMANT, MEMORY EFFICIENT AND SCALABLE MULTI-AGENT REINFORCEMENT LEARNING

## ABSTRACT

As the field of multi-agent reinforcement learning (MARL) progresses towards larger and more complex environments, achieving strong performance while maintaining memory efficiency and scalability to many agents becomes increasingly important. Although recent research has led to several advanced algorithms, to date, none fully address all of these key properties simultaneously. In this work, we introduce Sable, a novel and theoretically sound algorithm that adapts the retention mechanism from Retentive Networks to MARL. Sable's retention-based sequence modelling architecture allows for computationally efficient scaling to a large number of agents, as well as maintaining a long temporal context, making it well-suited for large-scale partially observable environments. Through extensive evaluations across six diverse environments, we demonstrate how Sable is able to significantly outperform existing state-of-the-art methods in the majority of tasks (34 out of 45, roughly 75%). Furthermore, Sable demonstrates stable performance as we scale the number of agents, handling environments with more than a thousand agents while exhibiting a linear increase in memory usage. Finally, we conduct ablation studies to isolate the source of Sable's performance gains and confirm its efficient computational memory usage. Our results highlight Sable's performance and efficiency, positioning it as a leading approach to MARL at scale.[1]

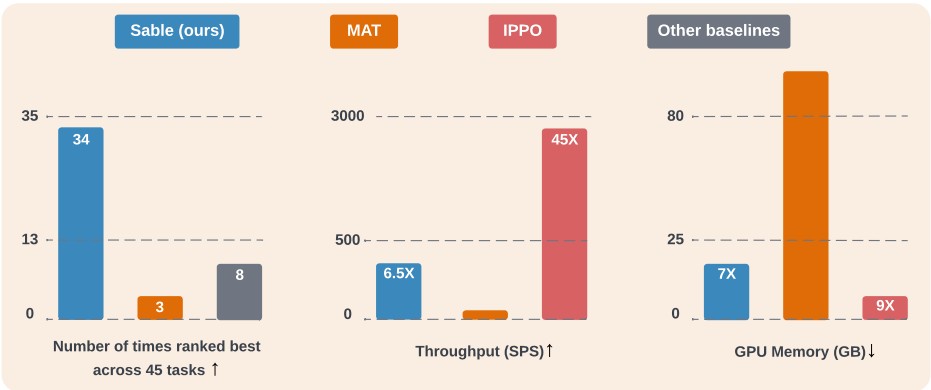

Figure 1: *Performance, memory, and scaling properties of **Sable**, aggregated over 45 cooperative MARL tasks.* **Left**: Sable ranks first in 34 out of 45 tasks, outperforming state-of-the-art MARL algorithms across 6 environments: RWARE (Papoudakis et al., 2021), LBF (Christianos et al., 2020), MABrax (Peng et al., 2021), SMAX (Rutherford et al., 2023), Connector (Bonnet et al., 2023) and MPE (Lowe et al., 2017). **Middle**: Sable exhibits superior throughput, processing up to 6.5 times more steps per second compared to the attention-based MAT (Wen et al., 2022) as we scale to 512 agents. **Right**: Sable scales efficiently to thousands of agents, maintaining stable performance, while using GPU memory as efficiently as independent systems, in this case IPPO (Witt et al., 2020), and significantly more efficiently than MAT.

---

[1]All experimental data and code is made available at: https://sites.google.com/view/sable-marl.

# 1 INTRODUCTION

When considering large-scale practical applications of multi-agent reinforcement learning (MARL) such as autonomous driving (Lian & Deshmukh, 2006; Zhou et al., 2021; Li et al., 2022) and electricity grid control (Kamboj et al., 2011; Li et al., 2016), it becomes increasingly important to maintain three key properties for a system to be effective: strong performance, memory efficiency, and scalability to many agents. Although many existing MARL approaches exhibit one or two of these properties, a solution effectively encompassing all three remains elusive.

To briefly illustrate our point, we consider the *spectrum of approaches to MARL* in terms of (1) performance (general ability to solve tasks at a moderate scale), (2) memory efficiency (the memory requirements to perform joint policy inference at execution time) and (3) scalability (the ability to maintain good performance as the number of agents grows large).

**Indepedent learning (IL)** — *memory efficient but not performant or scalable.* On the one end of the spectrum lies IL, or decentralised methods, where agents act and learn independently. As intuitively expected, the earliest work in MARL uses this approach (Tan, 1993; 1997). However, even early on, clear limitations were highlighted due to non-stationarity from the perspective of each learning agent (Claus & Boutilier, 1998), failing to solve even simple tasks. When deep neural networks were introduced into more modern MARL algorithms, these algorithms also followed the IL paradigm (Tampuu et al., 2017; Witt et al., 2020), with reasonable results. IL algorithms demonstrate proficiency in handling many agents in a memory efficient way by typically using shared parameters and conditioning on an agent identifier. However, at scale, the performance of IL methods remains suboptimal compared to more centralised approaches (Papoudakis et al., 2021; Yu et al., 2022; Wen et al., 2022).

**Centralised training with decentralised execution (CTDE)** — *performant and memory efficient but not yet scalable.* As a solution to the failings of independent learning and lying between decentralised and centralised methods, is CTDE (Kraemer & Banerjee, 2016). Here, centralisation improves learning by removing non-stationarity, while decentralised execution maintains memory efficient deployment. CTDE follows two main branches: value-based and actor-critic. In value-based methods, centralisation is achieved through a joint value function used during training which has a factorisation structure adhering to the individual-global-max (IGM) principle. This means that if each agent acts greedily at execution time it is equivalent to the team acting greedily according to the joint value function. Seminal work along this line include VDN (Sunehag et al., 2017) and QMIX (Rashid et al., 2018), with many followup works (Son et al., 2019; Rashid et al., 2020b; Wang et al., 2020a; Son et al., 2020; Yang et al., 2020; Rashid et al., 2020a). In actor-critic methods, a centralised critic is used during training, and at execution time, policies are deployed independently. Many popular single-agent actor-critic algorithms have CTDE MARL versions including MADDPG (Lowe et al., 2017), MAA2C (Papoudakis et al., 2020) and MAPPO (Yu et al., 2022), and have been combined with factorised critics (Wang et al., 2020b; Peng et al., 2021). Although CTDE helps during training to achieve better performance at execution time, centralised training may remain prohibitively expensive, especially if the size of the global state is agent dependent. Furthermore, independent policies, even when trained jointly, are often limited in their coordination capabilities when deployed at larger scale (Long et al., 2020; Christianos et al., 2021; Guresti & Ure, 2021).

**CTDE policy optimisation with theoretical guarantees** — *theoretically sound, performant and memory efficient but not yet scalable.* Until fairly recently, both value-based and actor-critic MARL had limited theoretical guarantees. This changed for actor-critic methods with a series of papers developing first trust region learning methods (Kuba et al., 2022a), and subsequently, *mirror learning* (Kuba et al., 2022b) for MARL, culminating in the *Fundamental Theorem of Heterogeneous-Agent Mirror Learning* (Kuba et al., 2022c) (Theorem 1). The theorem states that for specifically designed methods, that utilise a particular heterogeneous-agent update scheme during policy optimisation, monotonic improvement and convergence is guaranteed. Stemming from this work, a class of heterogeneous agent RL algorithms (Zhong et al., 2024) have been proposed including HATRPO, HAPPO, HAA2C, HADDPG and HASAC (Liu et al., 2023a). Although theoretically sound, these algorithms generally suffer from the same drawbacks as conventional CTDE methods in terms of practical performance at scale, for similar reasons (Guo et al., 2024)

**Centralised learning** — *theoretically sound with state-of-the-art performance but not yet memory efficient or scalable.* On the other end of the spectrum lie centralised algorithms. These include

classical RL algorithms that treat MARL as a single-agent problem with an expanded action space, as well as approaches that condition on global information during execution, e.g. graph-based communication methods (Zhu et al., 2022). A particularly interesting line of work has been to employ the use of transformers (Vaswani et al., 2017; Hu et al., 2021; Gallici et al., 2023; Yang et al., 2024) and re-frame MARL as a (typically offline) sequence modeling problem (Chen et al., 2021; Meng et al., 2021; Tseng et al., 2022; Zhang et al., 2022; Liu et al., 2023b; Forsberg et al., 2024). One such approach for *online* learning is the Multi-Agent Transformer (MAT) (Wen et al., 2022) which achieves state-of-the-art (SOTA) performance in cooperative MARL tasks. Although MAT is highly performant and theoretically sound (from Theorem 1 in Kuba et al. (2022c)), it has limitations: (1) MAT lacks the ability to scale to truly large multi-agent systems due to the inherent memory limitations of the attention mechanism (Katharopoulos et al., 2020), and (2) MAT lacks the ability to condition on observation histories. These limitations significantly impact what is possible to achieve at scale and in partially observable settings, commonly encountered in the real world.

**Our work** — *theoretically sound, state-of-the-art performance, memory efficient and scalable.* We seek to develop an approach capable of SOTA performance, while being memory efficient and able to scale to many agents. To achieve this, we take inspiration from the MAT architecture and recent work in linear recurrent models in RL (Lu et al., 2024; Morad et al., 2024b) to develop an online sequence modeling approach to MARL but one that is significantly more memory efficient and scalable. Our key innovation is to replace the attention mechanism in MAT with an RL adapted version of the *retention* mechanism used in the recently proposed Retentive Networks (RetNets) (Sun et al., 2023). We call our approach **Sable**. Sable has theoretical convergence guarantees, is able to handle settings with up to a thousand agents and can process entire episode sequences as memory, crucial for learning in partially observable settings. Through comprehensive benchmarks across 45 different tasks, we empirically verify that Sable significantly outperforms SOTA methods in the majority of cases (34 out of 45). This includes outperforming the SOTA fully centralised MAT, while being as memory efficient as fully decentralised IPPO, achieving the best of both sides of the MARL spectrum. We concretely summarise our contributions below:

- We develop (to the best of our knowledge) the first encoder-decoder RetNet that uses cross-retention. We further extend this encoder-decoder RetNet to have resettable hidden states over a temporal sequence to ensure that information does not flow across episode boundaries. This produces a RetNet-based architecture suitable for RL.

- We use the above innovations to build Sable which achieves SOTA performance, is able to reason over multiple timesteps, is memory efficient, scales to many agents and is by design theoretically sound. We believe Sable is the first demonstration of successfully using RetNets for learning policies in RL and more specifically, the first successful use of RetNets as a sequence modelling approach to online MARL. Sable provides the best trade-off in terms of memory efficiency when compared to independent learning, while significantly surpassing CTDE methods and MAT in terms of performance and scalability (Figure 1).

## 2   BACKGROUND

**Problem Formulation**   Cooperative MARL in partially observable settings can be modeled using a decentralised-POMDP with $\langle N, \mathcal{O}, \mathcal{A}, R, P, \gamma \rangle$. Here $N$ is the number of agents, $\mathcal{O} = \prod_{i=1}^{N} \mathcal{O}^i$ is the joint observation space of all agents, $\mathcal{A} = \prod_{i=1}^{N} \mathcal{A}^i$ is the joint action space of all agents, $R : \mathcal{O} \times \mathcal{A} \rightarrow \mathbb{R}$ is the joint reward function, $P : \mathcal{O} \times \mathcal{A} \times \mathcal{O} \rightarrow [0, 1]$ is the environment transition probability function and $\gamma \in [0, 1)$ is a discounting factor. At timestep $t$, each agent receives a separate observation $o_t^i \in \mathcal{O}^i$, collectively forming the joint observation $\boldsymbol{o}_t \in \mathcal{O}$, and executes a separate action $a_t^i \in \mathcal{A}^i$, forming the joint action $\boldsymbol{a}_t \in \mathcal{A}$, sampled from a joint policy $\boldsymbol{\pi}(\boldsymbol{a}|\boldsymbol{o}) = \prod_{i=1}^{N} \pi(a^i|o^i)$. All agents receive a shared reward $r_t = R(\boldsymbol{o}_t, \boldsymbol{a}_t)$. The goal is to learn an optimal joint policy which maximises the expected joint discounted reward $J = \mathbb{E}_{\boldsymbol{\pi}}[\sum_{t=0}^{\infty} \gamma^t r_t]$.

**Retention**   Retention as used in Retentive networks (RetNets) introduced by Sun et al. (2023), eliminates the softmax operator from attention and instead incorporates a time-decaying causal mask (decay matrix) with GroupNorm (Wu & He, 2018) and a swish gate (Hendrycks & Gimpel, 2016; Ramachandran et al., 2017) to retain non-linearity. This reformulation allows for the same computation to be expressed in three distinct but equivalent forms:

**1. Recurrent** which operates on a single input token at a time via a hidden state

$$h_s = \kappa h_{s-1} + K_s^{\mathrm{T}} V_s$$
$$\mathrm{Retention}(\boldsymbol{x}_s) = Q_s h_s, \quad s = 1, \ldots, S \tag{1}$$

where $Q_s, K_s, V_s$ are per token query, key and value matrices, respectively. Each of these is computed by applying learned projection matrices $W_Q$, $W_K$, and $W_V$ on the embedded input sequence $\boldsymbol{x}$. The decay factor $\kappa \in (0, 1)$ determines the rate at which information from earlier parts of the sequence is retained.

**2. Parallel** which operates on a batch of tokens in parallel akin to attention, given as

$$\mathrm{Retention}(\boldsymbol{x}) = (QK^T \odot D)V, \quad D_{sm} = \begin{cases} \kappa^{s-m}, & \text{if } s \geq m \\ 0, & \text{if } s < m, \end{cases} \tag{2}$$

where $D$ is referred to as the decay matrix.

**3. Chunkwise** which is a hybrid between the parallel and recurrent forms and allows for efficient long-sequence modeling. The approach involves splitting the sequence into $i$ smaller chunks, each of length $B$ and can be written as:

$$Q_{[i]} = Q_{B(i-1):Bi}, \quad K_{[i]} = K_{B(i-1):Bi}, \quad V_{[i]} = V_{B(i-1):Bi}$$
$$h_i = K_{[i]}^T (V_{[i]} \odot \zeta) + \kappa^B h_{i-1}, \quad \zeta_{ij} = \kappa^{B-i-1}$$
$$\mathrm{Retention}(\boldsymbol{x}_{[i]}) = (Q_{[i]} K_{[i]}^T \odot D)V_{[i]} + (Q_{[i]} h_{i-1}) \odot \xi, \quad \xi_{ij} = \kappa^{i+1}. \tag{3}$$

The above equivalent representations enable two key advantages over transformers: (1) it allows for constant memory usage during inference while still leveraging modern hardware accelerators for parallel training, and (2) it facilitates efficient handling of long sequences by using the chunkwise representation during training, which can be re-expressed in a recurrent form during inference.

## 3 METHOD

In this section, we introduce **Sable**, our approach to MARL as sequence modelling using a modified version of retention suitable for RL. Sable enables parallel training and memory-efficient execution at scale, with the ability to capture temporal dependencies across entire episodes. We explain how Sable operates during both training and execution, how we adapt retention to work in MARL, and provide different strategies for scaling depending on the problem setting.

**Execution** Sable interacts with the environment for a defined rollout length, $L$, before each training phase. During this interaction, the encoder uses a chunkwise representation, processing the observation of all agents at each timestep in parallel. A hidden state $h^{enc}$ maintains a memory of past observations and is reset at the end of each episode. During execution, the decay matrix, $D$, is set to all ones, allowing for full self-retention over all agents' observations within the same timestep. These adjustments to retention, particularly the absence of decay across agents and the resetting of memory at episode termination, result in the following encoder formulation during execution:

$$\mathrm{Retention}(\tilde{\boldsymbol{o}}_t) = Q_t h_t^{enc}, \quad t = l, \ldots, l + L$$
$$\text{where} \quad h_t^{enc} = \delta(\kappa h_{t-1}^{enc} + K_t^{\mathrm{T}} V_t), \quad \delta = \begin{cases} 0, & \text{if the episode has ended} \\ 1, & \text{if the episode is ongoing} \end{cases} \tag{4}$$

where $Q_t, K_t, V_t$ are query, key and value matrices of all agents' observations at timestep $t$ within the rollout that started at $l^{th}$ timestep and $\tilde{\boldsymbol{o}}_t$ is the embedded observation sequence.

The decoder operates recurrently over both agents and timesteps, decoding actions auto-regressively per timestep as follows:

$$\mathrm{Retention}(\tilde{a}_t^i) = Q_t^i \hat{h}_i,$$
$$\text{where} \quad \hat{h}_i = \hat{h}_{i-1} + (K_t^i)^{\mathrm{T}} V_t^i, \quad i = 1, \ldots, N, \tag{5}$$

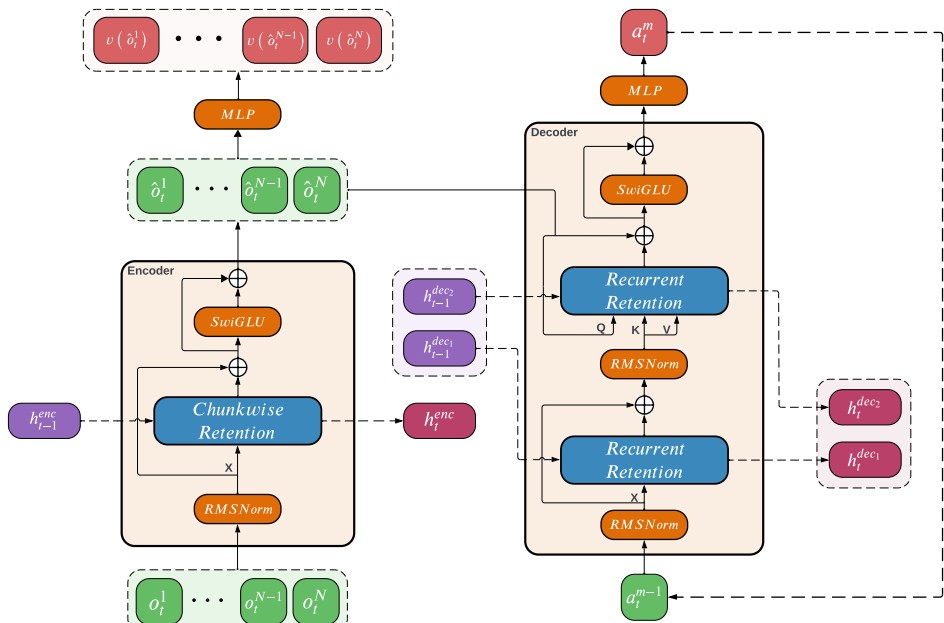

Figure 2: *Sable architecture and execution.* The encoder receives all agent observations $o_t^1, ..., o_t^N$ from the current timestep $t$ along with a hidden state $h_{t-1}^{enc}$ representing past timesteps and produces encoded observations $\hat{o}_t^1, ..., \hat{o}_t^N$, observation-values $v(\hat{o}_t^1), ..., v(\hat{o}_t^N)$ and a new hidden state $h_t^{enc}$. The decoder performs recurrent retention over the current action $a_t^{m-1}$, followed by cross attention with the encoded observations, producing the next action $a_t^m$. The initial hidden states for recurrence over agents in the decoder at the current timestep are $(h_{t-1}^{dec_1}, h_{t-1}^{dec_2})$ and by the end of the decoding process, it generates the updated hidden states $(h_t^{dec_1}, h_t^{dec_2})$.

with $\hat{h}_1 = h_{t-1}^{dec} + (K_t^1)^{\mathrm{T}} V_t^1$ and $h_t^{dec} = \delta(\kappa \hat{h}_N)$. Here, $Q_t^i, K_t^i, V_t^i$ are query, key and value matrices and $\tilde{a}_t^i$ the embedded action of the $i^{th}$ agent at timestep $t$. The hidden state $h^{dec}$ is carried across timesteps, decaying at the end of each timestep and resetting to zero when an episode ends. Within each timestep, the intermediate variable $\hat{h}_i$ is sequentially passed from one agent to the next and is used exclusively in the retention calculation. It is this auto-regressive action selection which leverages the advantage decomposition theorem to give Sable theoretically grounded convergence guarantees.

**Training** During training, Sable samples entire trajectories $\tau$ from an on-policy buffer and randomly permutes the order of agents within timesteps. The encoder takes as input a sequence of flattened agent-timestep observations from an entire trajectory: $[o_l^1, o_l^2, ..., o_{l+L}^{N-1}, o_{l+L}^N]$, representing a sequence of observations that start at timestep $l$. The decoder takes a similar sequence but of actions instead of observations as input. Sable uses the chunkwise representation for both encoding and decoding during training, allowing it to process entire trajectories in parallel while using a hidden state to maintain the memory of previous trajectories.

To implement the chunkwise formulation, Sable applies the decaying factor over time (not across agents), and resets the memory at the end of each episode through the $D$, $\zeta$ and $\xi$ matrices. This results in the following chunkwise training equation:

$$h_\tau = K_{[\tau]}^T (V_{[\tau]} \odot \zeta) + \delta \kappa^L h_{\tau_{prev}}, \quad \zeta = D_{NL, 1:NL}$$

$$\mathrm{Retention}(\boldsymbol{x}_{[\tau]}) = (Q_{[\tau]} K_{[\tau]}^T \odot D) V_{[\tau]} + (Q_{[\tau]} h_{\tau_{prev}}) \odot \xi, \quad (6)$$

$$\text{where} \quad \xi_{ij} = \begin{cases} \kappa^{\lfloor i/N \rfloor + 1}, & \text{if } i \le N t_{d_0} \\ 0, & \text{if } i > N t_{d_0} \end{cases}.$$

The floor operator in $\xi$, $\lfloor i/N \rfloor$, ensures that all agents from the same timestep share the same decay values. The input $\boldsymbol{x}_{[\tau]}$ is the sequence of observations from $\tau$ for the encoder and the sequence of actions for the decoder. We represent the index of the first terminal timestep in $\boldsymbol{x}_{[\tau]}$ as $t_{d_0}$ and use $h_\tau$ to denote the hidden state at the end of the current trajectory $\tau$. Finally, the matrix $D$ is a modified version of the decay matrix from standard retention, with dimensions $(NL, NL)$, which we define in more detail below.

In practice, rather than computing $h$ during training, we reuse the hidden states from the final step of the previous execution trajectory $\tau_{prev}$, which means that we replace $h_{\tau_{prev}}$ with $h_{l-1}^{enc}$ in the case of the encoder and $h_{l-1}^{dec}$ in the case of the decoder.

**Adapting the decay matrix for MARL**    In order for RetNets to work in RL, we make three key adaptations to the decay matrix used during training. First, we ensure that each agent's observations are decayed by the same amount within each timestep. Second, we ensure that the decay matrix accounts for episode termination so that information is not allowed to flow over episode boundaries. Third, we construct an agent block-wise decay matrix for the encoder to ensure that there is full self-retention over agents within each timestep. A more detailed discussion on the construction of the decay matrices as well as an illustrative example are given in Appendix E.

**Scaling and efficient memory usage**    In practical applications, there might be different axes of interest in terms of memory usage. For example, scaling the number of agents in the system, or efficiently handling the sequence length that can be processed at training time, i.e. the number of timesteps per update. We propose slightly different approaches for efficient memory use and scaling across each axis.

- *Scaling the number of agents.* Scaling to thousands of agents requires a significant amount of memory. Therefore, in this setting, we use MAT-style single-timestep sequences to optimise memory usage and reserve chunking to be applied across agents. This requires only changing the encoder during execution, as the decoder is already recurrent over both agents and timesteps. However, this change to the encoder makes it unable to perform full self-retention across agents, as it cannot be applied across chunks. During training, the process mirrors that of execution but is applied to both the encoder and decoder.

- *Scaling the trajectory context length.* Since the training sequence will grow proportional to $NL$ in the case where Sable maintains memory over trajectories, training could become computationally infeasible for tasks requiring long rollouts. In order to accommodate this, we chunk the flattened agent-timestep observation along the time axis during training with the constraint that agents from the same timestep must always be in the same chunk. This allows sable to process chunks of rollouts several factors smaller than the entire rollout length while maintaining a memory of the full sequence during processing.

**Theoretical monotonic improvement and convergence guarantees**    Sable inherits strong performance and convergence guarantees by design through its choice of using the Proximal Policy Optimisation (PPO) objective with autoregressive policy updates. These guarantees stem from recent theoretical results which we briefly outline here (for a more detailed discussion we refer the reader to the Appendix). First, is the advantage decomposition theorem/lemma (Kuba et al., 2021) (Lemma 1), which was used to develop trust region learning approaches for MARL with monotonic performance improvement guarantees (Kuba et al., 2022a) (Theorem 2). Even though PPO-style algorithms with autoregressive updates do not strictly adhere to trust region learning theory, and therefore, do not have strict monotonic improvement, more recent work has placed PPO within a class of *mirror learning* algorithms that indeed enjoy monotonic improvement and theoretical convergence guarantees (Kuba et al., 2022b) (Theorem 3.6). This work has since been extended to the multi-agent setting (Kuba et al., 2022c), (Theorem 1), and in particular, HAPPO (multi-agent PPO with heterogeneous autoregressive updates), was shown to be an instance of multi-agent mirror learning, and therefore theoretically sound. To obtain an instance of mirror learning requires defining a valid drift functional, neighbourhood operator and sampling distribution. In both MAT and Sable, these design choices are exactly as they are for HAPPO, and therefore, we claim that Sable inherits the same theoretical monotonic improvement and convergence guarantees as HAPPO, and by extension, MAT.

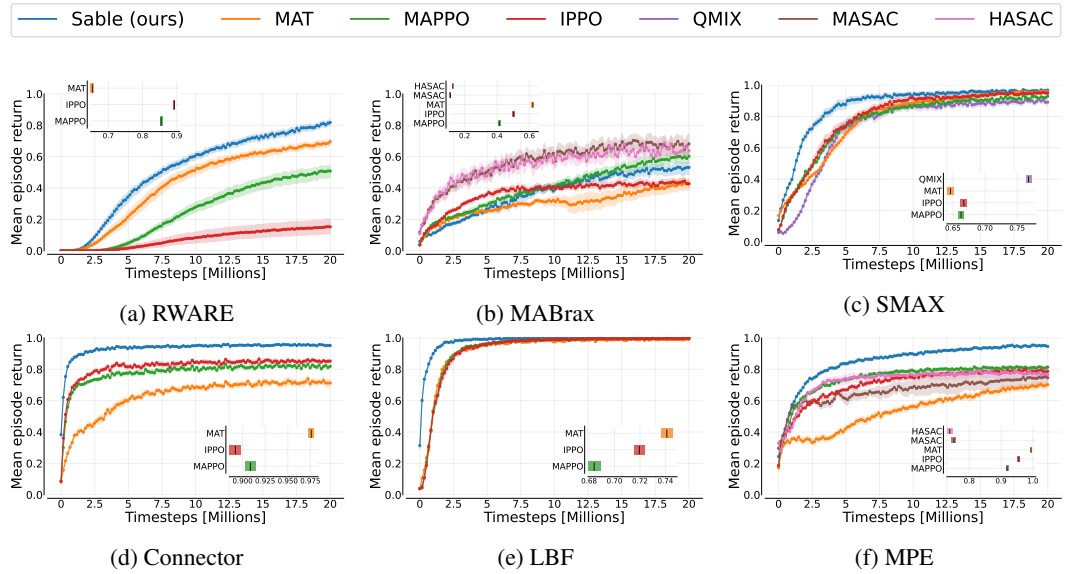

Figure 3: *Sample efficiency curves and probability of improvement scores aggregated per environment suite*. For each environment, results are aggregated over all tasks and the min-max normalised inter-quartile mean with 95% stratified bootstrap confidence intervals are shown. Inset plots indicate the overall aggregated probability of improvement for Sable compared to other baselines for that specific environment. A score of more than 0.5 where confidence intervals are also greater than 0.5 indicates statistically significant improvement over a baseline for a given environment (Agarwal et al., 2021).

**Code**  Our implementation of Sable is in JAX (Bradbury et al., 2023), and all code is available at: https://sites.google.com/view/sable-marl.

## 4  EXPERIMENTS

We validate the performance, memory efficiency and scalability of Sable by comparing it against several SOTA baseline algorithms from the literature. These baselines can broadly be divided into two groups. The first group consists of heterogeneous agent algorithms that leverage the advantage decomposition theorem. To the best of our knowledge, the Multi-Agent Transformer (MAT) (Wen et al., 2022) represents the current SOTA for cooperative MARL on discrete environments, and Heterogeneous Agent Soft Actor-Critic (HASAC) (Liu et al., 2023a) the current SOTA on continuous environments. The second group includes well-established baseline algorithms, including IPPO (Witt et al., 2020), MAPPO (Yu et al., 2022), QMIX (Rashid et al., 2020a) and MASAC. For all baselines, we use the JAX-based MARL library Mava (de Kock et al., 2023).

**Evaluation protocol**  We train each algorithm for 10 independent trials for each task. Each training run is allowed 20 million environment timesteps with 122 evenly spaced evaluation intervals. At each evaluation, we record the mean episode return over 32 episodes and, where relevant, any additional environment specific metrics (e.g. win rates). In line with the recommendations of Gorsane et al. (2022), we also record the absolute performance. For task-level aggregation, we report the mean with 95% confidence intervals while for aggregations over entire environment suites, we report the min-max normalised inter-quartile mean with 95% stratified bootstrap confidence intervals. Following from Agarwal et al. (2021), we consider algorithm $X$ to have significant improvement over algorithm $Y$ if the probability of improvement score and all its associated confidence interval values are greater than 0.5. All our evaluation aggregations, metric calculations and plotting leverages the MARL-eval library from Gorsane et al. (2022).

**Environments**  We evaluate Sable on several JAX-based benchmark environments including Robotic Warehouse (RWARE) (Papoudakis et al., 2021), Level-based foraging (LBF) (Christianos et al., 2020),

Table 1: *Per environment episode return.* Inter-quartile mean of the absolute episode returns with 95% stratified bootstrap confidence intervals. Bold values indicate the highest score per environment and an asterisk indicates that a score overlaps with the highest score within one confidence interval.

| Environment | Sable (Ours) | MAT | MAPPO | IPPO | MASAC | HASAC | QMIX |
|---|---|---|---|---|---|---|---|
| RWARE | $\mathbf{0.81}_{(0.79,0.83)}$ | $0.69_{(0.67,0.71)}$ | $0.51_{(0.47,0.54)}$ | $0.15_{(0.11,0.2)}$ | / | / | / |
| MABrax | $0.56_{(0.53,0.58)}$ | $0.45_{(0.42,0.47)}$ | $0.6_{(0.57,0.64)}$ | $0.5_{(0.49,0.52)}$ | $\mathbf{0.82}_{(0.77,0.86)}$ | $0.8^{*}_{(0.76,0.83)}$ | / |
| SMAX | $\mathbf{0.94}_{(0.92,0.95)}$ | $0.92^{*}_{(0.91,0.94)}$ | $0.86_{(0.84,0.87)}$ | $0.93^{*}_{(0.91,0.94)}$ | / | / | $/\ 0.86_{(0.84,0.88)}$ |
| Connector | $\mathbf{0.95}_{(0.95,0.95)}$ | $0.88_{(0.88,0.89)}$ | $0.91_{(0.91,0.92)}$ | $0.93_{(0.92,0.93)}$ | / | / | / |
| LBF | $\mathbf{1.0}_{(1.0,1.0)}$ | $0.99_{(0.98,0.99)}$ | $\mathbf{1.0}_{(1.0,1.0)}$ | $0.99^{*}_{(0.99,1.0)}$ | / | / | / |
| MPE | $\mathbf{0.95}_{(0.94,0.95)}$ | $0.69_{(0.68,0.71)}$ | $0.81_{(0.8,0.81)}$ | $0.79_{(0.78,0.8)}$ | $0.76_{(0.72,0.8)}$ | $0.79_{(0.76,0.82)}$ | / |

Connector (Bonnet et al., 2023), The StarCraft Multi-Agent Challenge in JAX (SMAX) (Rutherford et al., 2023), Multi-agent Brax (MABrax) (Peng et al., 2021) and the Multi-agent Particle Environment (MPE) (Lowe et al., 2017). All environments have discrete action spaces with dense rewards, except for MABrax and MPE, which have continuous action spaces and RWARE which has sparse rewards. Furthermore, we compare to HASAC and MASAC only on continuous tasks given their superiority in this setting and QMIX only on SMAX as it has been shown to perform suboptimally in other discrete environments (most notably in spare reward settings such as RWARE) (Papoudakis et al., 2020). Finally, we highlight that our evaluation suite comprised of 45 tasks represents nearly double the amount of tasks used by prior benchmarking work (Papoudakis et al., 2020) and substantially more than conventional research work recently published in MARL (Gorsane et al., 2022).

**Hyperparameters** All baseline algorithms as well as Sable were tuned on each task with a tuning budget of 40 trials using the Tree-structured Parzen Estimator (TPE) Bayesian optimisation algorithm from the Optuna library (Akiba et al., 2019). For a discussion on how to access all task hyperparameters and for all tuning search spaces, we refer the reader to Appendix D.

## 4.1 PERFORMANCE

In Figure 1, we report the amount of times that an algorithm had a significant probability of improvement over all other algorithms on a given task. Furthermore, we present the per environment aggregated sample efficiency curves, probability of improvement scores and episode returns in Figure 3 and Table 1. Our experimental evidence shows Sable achieving SOTA performance across a wide range of tasks. Specifically, Sable exceeds baseline performance on 34 out of 45 tasks. The only environment where this is not the case is on MABrax. For continuous robotic control tasks SAC is a particularly strong baseline, typically outperforming on-policy methods such as PPO (Haarnoja et al., 2018; Huang et al., 2024; Freeman et al., 2021a). Given that Sable uses the PPO objective for training, this performance is unsurprising. However, Sable still manages to achieve SOTA performance in continuous control tasks on MPE. We note that previous benchmarking and evaluation work (Papoudakis et al., 2020; Gorsane et al., 2022) has recommended training off-policy algorithms for a factor of 10 less environment interactions than on-policy algorithms due to more gradient updates for the same number of environment interactions. In our case, we find that off-policy systems do roughly 15 times more gradient updates for the same amount of environment interactions. If we had done this the performance of HASAC, MASAC and QMIX would have been less performant than reported here. Additional tabular results, task and environment level aggregated plots are given in Appendix C.

## 4.2 MEMORY USAGE AND SCALABILITY

We assess Sable's ability to efficiently utilise computational memory, focusing primarily on scaling across the agent axis.

**Challenges in testing scalability using standard environments** Testing scalability and memory efficiency in standard MARL environments poses challenges, as many environments such as SMAX, MPE and Connector, expand the observation space as the number of agents grows. MABrax has uniquely assigned roles per agent making it difficult to scale up and RWARE does not have a straightforward way to ensure task difficulty as the number of agents increases. For these reasons, the above environments are difficult to use when testing algorithmic scalability without significantly

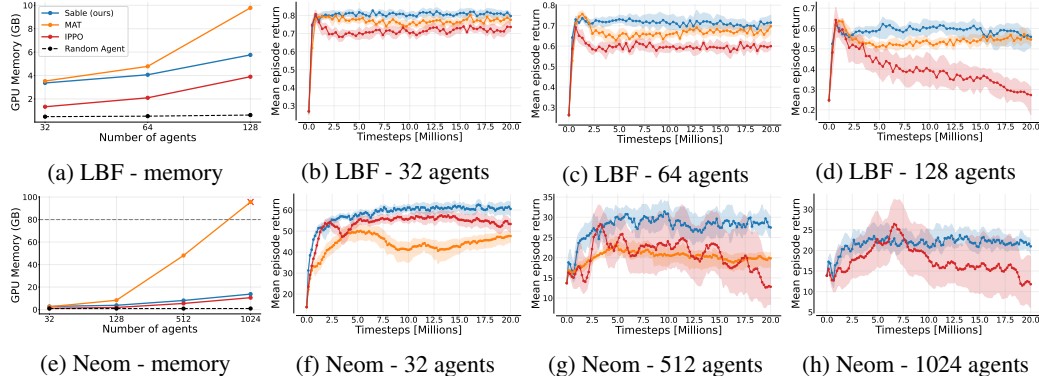

(a) LBF - memory     (b) LBF - 32 agents     (c) LBF - 64 agents     (d) LBF - 128 agents

(e) Neom - memory     (f) Neom - 32 agents     (g) Neom - 512 agents     (h) Neom - 1024 agents

Figure 4: *Memory usage and agent scalability.* When scaling to many agents, Sable is able to achieve superior converged performance while maintaining memory efficiency.

modifying the original environment code. Among these, LBF is unique because it is easier to adjust by reducing agents' field of view (FOV) while maintaining reasonable state size and offering faster training. However, it still requires modifications to ensure a fixed observation size beside the FOV (see Appendix B.3 for more details). Despite these adjustments, LBF could not fully demonstrate Sable's scaling capability, as it could not scale past 128 agents due to becoming prohibitively slow. Therefore, to explore scaling up to a thousand agents, we introduce **Neom**, a fully cooperative environment specifically designed to test algorithms on larger numbers of agents.

**A new environment for testing agent scalability in cooperative MARL** A task in Neom is characterised by a periodic, discretised 1-dimensional pattern that is repeated for a given number of agents. Each agent observes whether it is in the correct position and the previous actions it has taken. Agents receive a shared reward which is calculated as the Manhattan distance between the team's current pattern and the underlying task pattern. We design three task patterns: (1) `simple-sine`: $\{0.5, 0.7, 0.8, 0.7, 0.5, 0.3, 0.2, 0.3\}$, (2) `half-1-half-0`: $\{1, 0\}$, (3) `quick-flip`: $\{0.5, 0, -0.5, 0\}$. For more details, see Appendix B.7.

**Experimental setup** We evaluate the performance of Sable, MAT, and IPPO on the LBF and Neom environments. For LBF, experiments involve tasks with 32, 64, and 128 agents, while for Neom we include tasks with 32, 512 and 1024 agents. Given the slower throughput of these tasks, hyperparameter tuning was not feasible. Instead, we take the optimal hyperparameters found on reasonably sized tasks and apply these across all larger tasks. For example, for LBF we use the `15x15-3p-5f` task as it appears to be one of the hardest (See Appendix Figure 11) and for Neom we use `simple-sine-16-ag`, `half-1-half-0-16-ag` and `quick-flip-16-ag` as reference tasks. To measure the memory usage efficiency on both LBF and Neom environments on the agents' axis, we select 32 as the fixed chunk size, which corresponds to the smallest number of agents used in our experiments.

**Results** We observe the following from the results in Figure 4. First, although IPPO scales well from a memory perspective, it is unable to learn with many agents. It achieves a high reward initially but its performance degrades as training continues. Second, we find that Sable can consistently outperform MAT. Although the margin is small, this performance is achieved while maintaining comparable computational memory usage to IPPO, whereas MAT scales poorly and requires more than the maximum available GPU memory (80GB) on Neom tasks with 1024 agents. Notably, for Neom with 1024 agents, Sable sustains a stable mean episode return of around 25, indicating that approximately 40% to 50% of the agents successfully reached the target location. This result is significant given the shared reward structure, which poses a difficult coordination challenge for such a large population of agents.

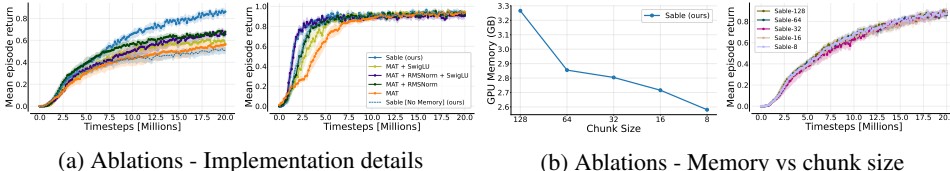

(a) Ablations - Implementation details     (b) Ablations - Memory vs chunk size

Figure 5: *Ablation studies on RWARE and SMAX.* **(a)** Comparing Sable with MAT with modifications from Sable's implementation details. **(b)** Showing the relationship between chunk size, performance and memory usage on RWARE.

### 4.3 ABLATIONS

We aim to better understand the source of Sable's performance gains compared to MAT. There are two specific implementation details that Sable inherits from the retention mechanism that can easily be transferred to attention, and therefore to MAT. The first is using root mean square normalization (RMSNorm) (Zhang & Sennrich, 2019) instead of layer normalization (Lei Ba et al., 2016) and the second is using SwiGLU layers (Shazeer, 2020; Ramachandran et al., 2017) instead of feed forward layers. Other than implementation details, the difference between MAT and Sable is that Sable uses retention instead of attention and it conditions on histories instead of on single timesteps. To determine the reason for the performance difference between Sable and MAT, we adapt MAT to use the above implementation changes, both independently and simultaneously and we compare MAT to a version of Sable with no memory that only conditions on the current timestep. We test all three variants of MAT and the Sable variant on two RWARE tasks (`tiny-4ag` and `medium-4ag`) and two SMAX tasks (`3s5z` and `smacv2_5_units`) and compare them with the original implementation. We tune all methods using the same protocol as the main results.

In Figure 5a, we see that the above implementation details do make a difference to MAT's performance. In RWARE, MAT's variants slightly increase in both performance and sample efficiency. However, Sable still achieves significantly higher performance while maintaining a similar sample efficiency. The same cannot be said for Sable without memory which performs similarly to the default MAT and significantly worse than MAT with the implementation improvements. In SMAX, we observe a marked increase in sample efficiency for MAT equalling the sample efficiency of Sable and outperforming Sable's sample efficiency without memory, but no increase in overall performance. This is likely due to the fact that both MAT and Sable already achieve close to the maximum performance in these SMAX environments. In summary, we find that these implementation details do matter and improve the performance and sample efficiency of MAT, although not to the level of Sable's performance. When we compared MAT, Sable, and Sable without memory, we discovered that Sable's performance improvement stems from its ability to use temporal memory, rather than from the retention mechanism itself. This is evident because Sable without memory (which performs similarly to MAT) differs from Sable only in how the input sequence is structured.

In Figure 5b, we see that even when dividing the rollout trajectories into chunks that are up to a factor of 16 smaller than the full rollout length, Sable's performance remains consistent, while its memory usage decreases.

## 5 CONCLUSION

In this work, we introduced Sable, a novel cooperative MARL algorithm that employs retentive networks to achieve significant advancements in memory efficiency, agent scalability and performance. Sable's ability to condition on entire episodes provides it with an enhanced temporal awareness, leading to SOTA performance. This is evidenced by our extensive evaluation, where Sable significantly outperforms other leading approaches in 75% of tasks tested. Moreover, Sable's memory efficiency complements its performance by addressing the significant challenge of scaling MARL algorithms as it is able to maintain stable performance even when scaled to over 1000 agents. Looking ahead, we aim to explore Sable's integration into more complex, larger-scale, real-world environments.

## 6 REPRODUCIBILITY STATEMENT

We have explained our hyperparameter tuning procedure and outlined all search spaces and default hyperparameters for all algorithms in Appendix D. We make all our code and raw experiment data available. Along with our code, we include all final hyperparameter values and scripts to relaunch all training runs. Aside from what we make available, our code is written in JAX which supports manual random state handling, this should make training runs more reproducible.

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

# APPENDIX

## A  SEQUENCE MODELLING RELATED WORK

**Linear recurrent models** Recent work in RL has leveraged structured state space models (Lu et al., 2024) for efficient long context memory. It has also been shown that various linear recurrent models including Linear Transformers (Katharopoulos et al., 2020), Fast and Forgetful Memory (Morad et al., 2024a), and Linear Recurrent Units (Orvieto et al., 2023) can be used for temporal memory in RL (Morad et al., 2024b). Sable falls into this category of algorithms due to leveraging the RetNet architecture, a linear recurrent model, instead of the Transformer.

**Transformers and RetNets in reinforcement learning** Other works have applied Transformers in the context of MARL (Hu et al., 2021; Wen et al., 2022). The closest to our work is MAT (Wen et al., 2022). In single-agent RL, transformers have been used to enable long range memory (Parisotto et al., 2020; Esslinger et al., 2022), most notably the Gated Transformer-XL (Parisotto et al., 2020). Sable differs from these works for two main reasons: (1) it is a distinctly multi-agent algorithm and (2) it has no need for appending observation histories to input sequences since it can retain all necessary information from previous timesteps with a hidden state. Moreover, and to the best of our knowledge, Sable is the first architecture to leverage RetNets for learning policies in RL. The only other application of RetNets has been to learn an efficient world model (Cohen et al., 2024).

## B  ENVIRONMENT DETAILS

### B.1  ROBOT WAREHOUSE

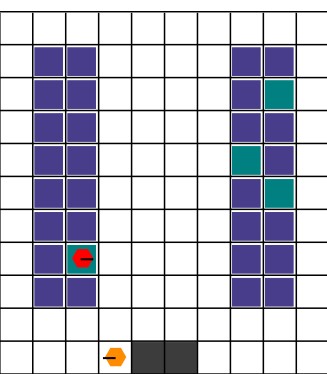

Figure 6: Environment rendering for Robot Warehouse. Task name: `tiny-2ag`.

The Robot Warehouse (RWARE) environment simulates a warehouse where robots autonomously navigate, fetching and delivering requested goods from specific shelves to workstations and then returning them. Inspired by real-world autonomous delivery depots, the goal in RWARE is for a team of robots to deliver as many randomly placed items as possible within a given time budget.

The version used in this paper is a JAX-based implementation of the original RWARE environment (Papoudakis et al., 2021) from the Jumanji environment suite (Bonnet et al., 2023). For this reason, there is a minor difference in how collisions are handled. The original implementation has some logic to resolve collisions, whereas the Jumanji implementation simply ends an episode if two agents collide.

**Naming convention**  The tasks in the RWARE environment are named according to the following convention:

<size>-<num_agents>ag<diff>

Each field in the naming convention has specific options:

- `<size>`: Represents the size of the Warehouse which defines the number of rows and columns of groups of shelves within the warehouse (e.g. tiny, small, medium, large).

- `<num_agents>`: Indicates the number of agents.

- `<diff>`: Optional field indicating the difficulty of the task, where 'easy' and 'hard' imply $2N$ and $N/2$ requests (shelves to deliver) respectively, with $N$ being the number of agents. The default is to have $N$ requests.

In this environment, we introduced an extra grid size named "xlarge" which expands the default "large" size. Specifically, it increases the number of rows in groups of shelves from three to four, while maintaining the same number of columns.

**Observation space**  In this environment observation are limited to partial visibility where agents can only perceive their surroundings within a 3x3 square grid centred on their position. Within this area, agents have access to detailed information including their position and orientation, as well as the positions and orientations of other agents. Additionally, they can observe shelves and determine whether these shelves contain a package for delivery.

**Action space**  The action space is discrete and consists of five total actions that allow for navigation within the grid and delivering the requested shelves. These actions include no operation (stop), turning left, turning right, moving forward, and either loading or unloading a shelf.

**Reward**  Agents receive a reward of 1 for each successful delivery of a requested shelf, coloured in green in Figure 6, to a designated goal (in black) and 0 otherwise. Achieving this reward demands a sequence of successful actions, making it notably sparse.

## B.2   SMAX

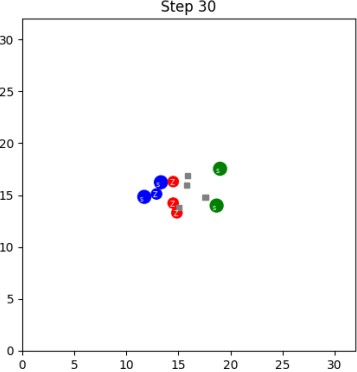

Figure 7: Environment rendering for SMAX. Task name: 2s3z.

SMAX, introduced by Rutherford et al. (2023), is a re-implementation of the StarCraft Multi-agent Challenge (SMAC) (Samvelyan et al., 2019) environment using JAX for improved computational efficiency. This redesign eliminates the necessity of running the StarCraft II game engine, thus results on this environment are not directly comparable to results on original SMAC. In this environment, agents collaborate in teams composed of diverse units to win the real-time strategy game StarCraft. For an in-depth understanding of the environment's mechanics, we refer the reader to the original paper (Samvelyan et al., 2019).

**Observation space**  Each agent observes all allies and enemies within its field of view, including itself. The observed attributes include position, health, unit type, weapon cooldown, and previous action.

**Action space**    Discrete action space that includes 5 movement actions: four cardinal directions, a stop action, and a shoot action for each visible enemy.

**Reward**    In SMAX, unlike SMAC, the reward system is designed to equally incentivise tactical combat and overall victory. Agents earn $50\%$ of their total return from hitting enemies and the other $50\%$ from winning the episode which ensures that immediate actions and ultimate success are equally important.

## B.3    Level Based Foraging

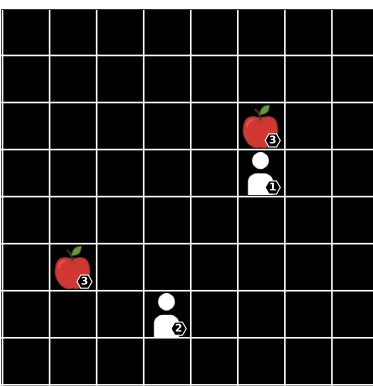

Figure 8: Environment rendering for Level-Based Foraging. Task name: `2s-8x8-2p-2f`.

In the Level-Based Foraging environment (LBF) agents are assigned different levels and navigate a grid world where the goal is to collect food items by cooperating with other agents if required. Agents can only consume food if the combined level of the agents adjacent to a given item of food exceeds the level of the food item. Agents are awarded points when food is collected.

The version used in the paper is a JAX-based implementation of the original LBF environment (Christianos et al., 2020) from the Jumanji environment suite (Bonnet et al., 2023). To the best of our knowledge, there are no differences between Jumanji's implementation and the original implementation.

**Naming convention**    The tasks in the LBF environment are named according to the following convention:

<obs>-<x_size>x<y_size>-<n_agents>p-<food>f<force_c>

Each field in the naming convention has specific options:

- <obs>: Denotes the field of view (FOV) for all agents. If not specified, the agents can see the full grid.
- <x_size>: Size of the grid along the horizontal axis.
- <y_size>: Size of the grid along the vertical axis.
- <n_agents>: Number of agents.
- <food>: Number of food items.
- <force_c>: Optional field indicating a forced cooperative task. In this mode, the levels of all the food items are intentionally set equal to the sum of the levels of all the agents involved. This implies that the successful acquisition of a food item requires a high degree of cooperation between the agents since no agent is able to collect a food item by itself.

**Observation space**    As shown in Figure 8, the 8x8 grid includes 2 agents and 2 foods. In this case, the agent has a limited FOV labelled "2s", indicating a 5x5 grid centred on itself where it can only observe the positions and levels of the items in its sight range.

**Action space**    The action space in the LBF is discrete, comprising six actions: no-operation (stop), picking up a food item (apple), and movements in the four cardinal directions (left, right, up, down).

**Reward**    The reward is equal to the sum of the levels of collected food divided by the level of the agents that collected them.

**Adapting the LBF environment for scalability experiments**    In the original Level-Based Foraging (LBF) implementation, agents processed complete grid information, including items outside their FOV, managed by masking non-visible items with the placeholder (-1, -1, 0) where each element of this triplet stands for (x, y, level). To improve computational efficiency, we revised the implementation to completely remove non-visible elements from the observation data, significantly reducing the observation size and ensuring agents process only relevant information within their FOV.

For instance, with a standardised FOV of 2, as illustrated in Figure 8, an agent sees a 5x5 grid centred around itself. Non-visible items are now excluded from the observation array which makes it easy to convert the agent's vector observation with level one from `[1, 2, 3, -1, -1, 0, 2, 2, 1, -1, -1, 0]` to `[ 1, 2, 3, 2, 2, 1]`. However, in tasks with numerous interacting agents, where the dynamics of visible items consistently change, fixed array sizes are required. We address this by defining the maximum number of visible items as $(2 \times \text{FOV} + 1)^2$, filling any excess with masked triplets to keep uniform array dimensions.

We designed three scenarios to test scalability on LBF using the standardised FOV of 2, ensuring a maximum of 25 visible items within the 5x5 grid, including each agent's information. To manage agent density as the environment scales, we created the following configurations for our experiments: `2s-32x32-32p-16f`, `2s-45x45-64p-32f`, and `2s-64x64-128p-64f`.

### B.4    CONNECTOR

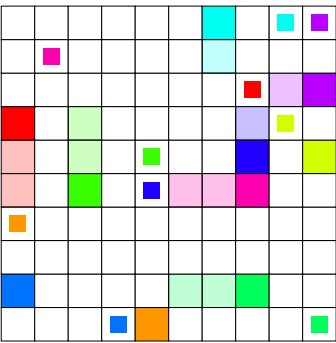

Figure 9: Environment rendering for Connector. Task name: `con-10x10-10a`.

The Connector environment consists of multiple agents spawned randomly into a grid world with each agent representing a start and end position that needs to be connected. The goal of the environment is to connect each start and end position in as few steps as possible. However, when an agent moves it leaves behind a path, which is impassable by all agents. Thus, agents need to cooperate to allow the team to connect to their targets without blocking other agents.

**Naming convention**    In our work, we follow this naming convention for the Connector tasks:



`con-<x_size>x<y_size>-<num_agents>a`



Each field in the naming convention means:

- `<x_size>`: Size of the grid along the horizontal axis.
- `<y_size>`: Size of the grid along the vertical axis.
- `<num_agents>`: Indicates the number of agents.

**Observation space**   All agents view an $n \times n$ square centred around their current location, within their field of view they can see trails left by other agents along with the target locations of all agents. They also observe their current $(x, y)$ position and their target's $(x, y)$ position.

**Action space**   The action space is discrete, consisting of five movement actions within the grid world: up, down, left, right, and no-operation (stop).

**Reward**   Agents receive $+1$ on the step where they connect and $-0.03$ otherwise. No reward is given after connecting.

### B.5   MUTLI-AGENT PARTICLE ENVIRONMENTS

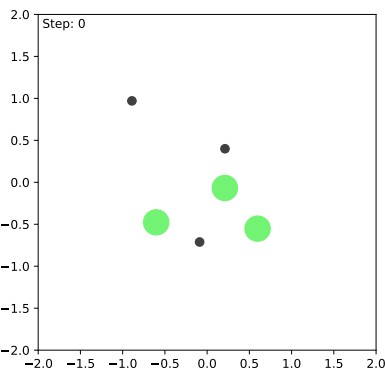

Figure 10: Environment rendering for Mutli-Agent Particle. Task name: `simple_spread_3ag`.

The Multi-Agent Particle Environments (MPE) comprises physics-based environments within a 2D world, where particles (agents) move, interact with fixed landmarks, and communicate. We focus exclusively on the "simple-spread" tasks, the only non-communication, non-adversarial setting in the suite where agents cooperate instead. In this setting, agents aim to cover landmarks to gain positive rewards and avoid collisions, which result in penalties. We employ a JAX-based clone of the original environment from the JaxMARL suite (Rutherford et al., 2023).

Contrary to the suggestions made by Gorsane et al. (2022), we only experiment on the `simple-spread` for the previous reasons, thus, we go beyond only the recommended version of `simple-spread` (`simple-spread-3ag`) and also test on 5 and 10 agents variants.

**Naming convention**   In our case, the `simple-spread` tasks used in the MPE environment are named according to the following convention:

$$\texttt{simple\_spread\_<num\_agents>ag}$$

`<num_agents>`: Indicates the number of agents in the simple spread task where we set the number of landmarks equal to the number of agents.

**Observation space**   Agents observe their own position and velocity as well as other agents positions and landmark positions.

**Action space**   Continuous actions space with 4 actions. Each action represents the velocity in all cardinal directions.

**Reward**   Agents are rewarded based on how far the closest agent is to each landmark and receive a penalty if they collide with other agents.

## B.6 MABRAX

MaBrax (Rutherford et al., 2023) is an implementation of the MaMuJoCo environment (Peng et al., 2021) in JAX, from the JAXMARL repository. The difference is that it uses BRAX (Freeman et al., 2021b) as the underlying physics engine instead of MuJoCo. Both MaMuJoCo and MaBrax are continuous control robotic environment, where the robots are split up so that certain joints are controlled by different agents. For example in *ant_4x2* each agent controls a different leg of the ant. The splitting of joints is the same in both MaBrax and MaMuJoCo.

The goal is to move the agent forward as far and as fast as possible. The reward is based on how far the agent moved and how much energy it took for the agent to move forward.

**Observation space**    Observations are separated into local and global observations. Globally, all agents observe the position and velocity of the root body. Locally agents observe the position and velocity of their joints as well as the position and velocity of their neighboring joints.

**Action space**    A continuous space where each agent controls some number of joints $n$. Each of the $n$ actions are bounded in the range $[-1, 1]$ and the value controls the torque applied to a corresponding joint.

**Reward**    Agents receive the reward from the single agent version of the environment. Positive reward is given if the agent moves *forward* and negative reward is given when energy is used to move the joints. Thus, agents are incentivised to move forward as efficiently as possible.

## B.7 NEOM

Neom tasks require agents to match a periodic, discretised 1D pattern that is repeated across the given number of agents. These tasks are specifically designed to assess the agents' ability to synchronise and reproduce specified patterns in a coordinated manner and in a limited time frame.

**Naming convention**    The tasks in the Neom environment are named according to the following convention:

<pattern-type>-<num_agents>ag

Each field in the naming convention has specific options:

- <pattern-type>: Represents the selected pattern for the agents to create ( "simple-sine", "half-1-half-0", and "quick-flip").
- <num_agents>: Indicates the number of agents.

**Observation space**    The observation space consists of a binary indicator showing whether the agent is in the correct position, concatenated with the agent's previous actions.

**Action space**  The action space consists of unique elements in the pattern, with each element defining an actions:

- `simple-sine`: $\{0.2, 0.3, 0.5, 0.7, 0.8\}$
- `half-1-half-0`: $\{1, 0\}$
- `quick-flip`: $\{0.5, 0, \text{-}0.5\}$

**Reward**  The reward function is calculated using the mean Manhattan distance between the team's current pattern and the target pattern. The reward ranges from 1 for a perfect match to -1 for the maximum difference, with normalization applied. Additionally, if the pattern is correct, the agents receive a bonus that starts at a maximum value of 9.0 and gradually decreases as the episode progresses, based on how much time has passed.

## C  FURTHER EXPERIMENTAL RESULTS

### C.1  ADDITIONAL PER TASK AND PER ENVIRONMENT RESULTS

In Figure 11, we give all task-level aggregated results. In all cases, we report the mean with 95% bootstrap confidence intervals over 10 independent runs. In Figure 12, we give the performance profiles for all environment suites.

### C.2  ADDITIONAL TABULAR RESULTS

When reporting tabular results, it can be challenging to represent information from an entire training run as a single value for a given independent trial. For this reason we give all tabular results for different aggregations. In all cases here, the aggregation method we refer to is the method that was used to aggregate a given training run into a point estimate. For aggregation over these point estimates we always compute the mean over independent trials along with the 95% bootstrap confidence interval.

#### C.2.1  MEAN OVER THE FULL TIMESERIES

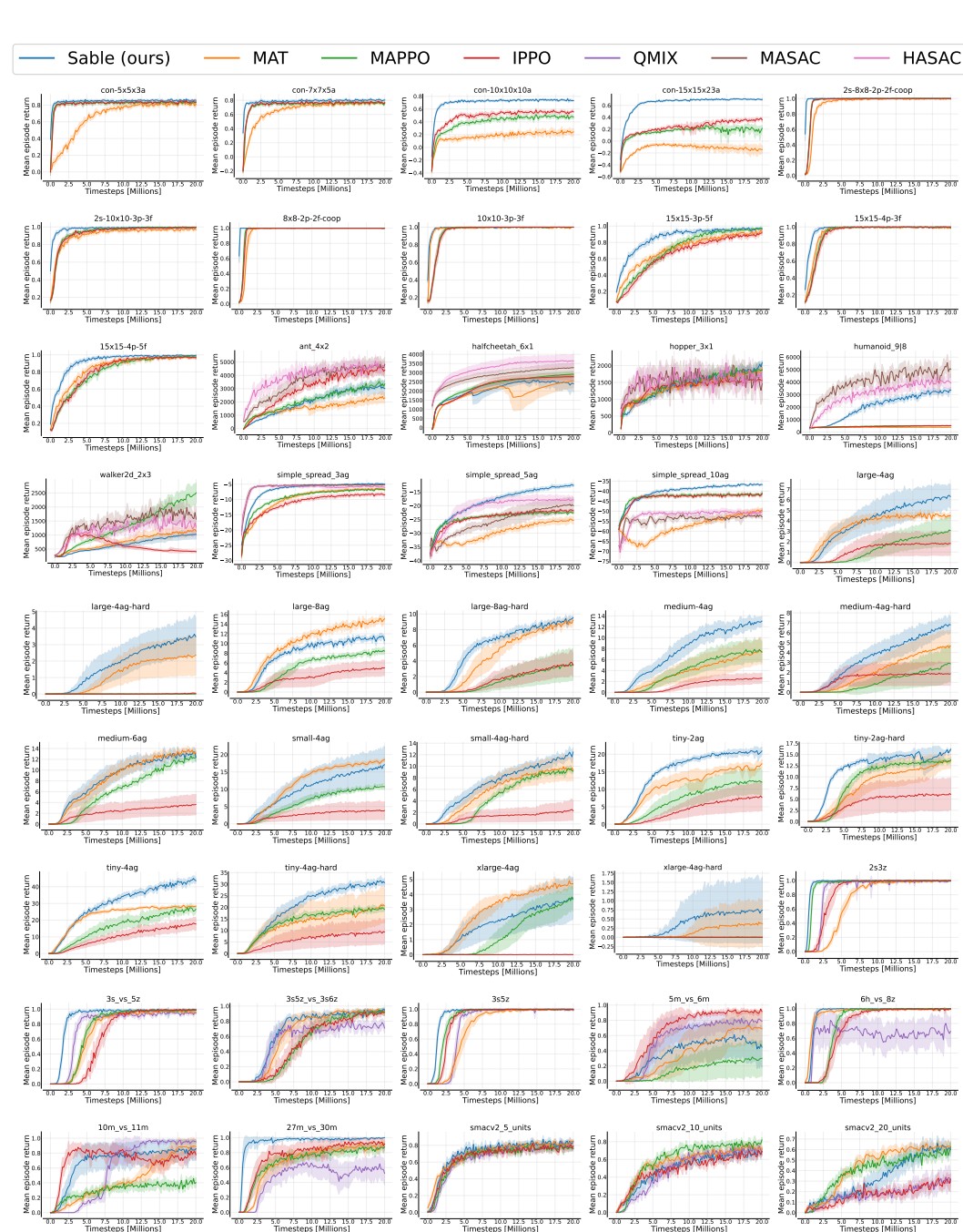

Figure 11: Mean episode return with 95% bootstrap confidence intervals on all tasks.

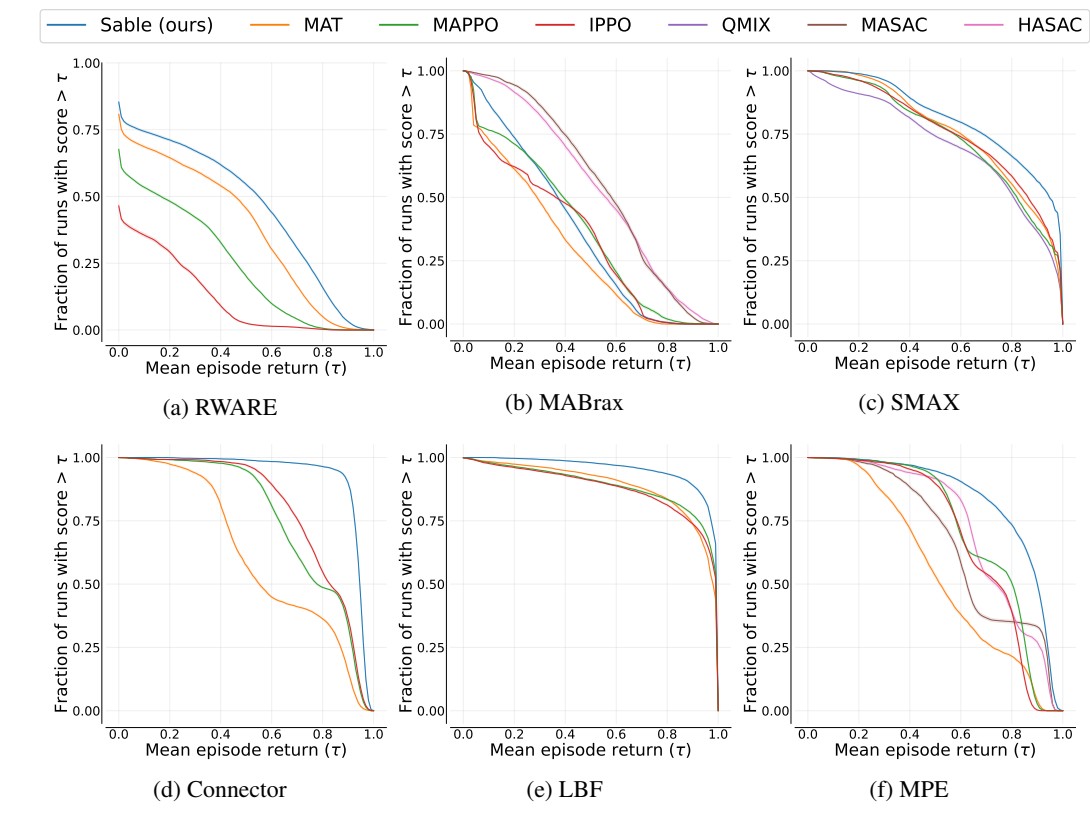

Figure 12: Per environment performance profiles.

Table 2: Mean episode return over training with 95% bootstrap confidence intervals for all tasks. Bold values indicate the highest score per task and an asterisk indicates that a score overlaps with the highest score within one confidence interval.

| | Task | Sable (Ours) | MAT | MAPPO | IPPO | MASAC | HASAC | QMIX |
|---|---|---|---|---|---|---|---|---|
| **Rware** | tiny-2ag | **15.33**(14.91,15.69) | 11.32(9.93,12.47) | 7.20(4.83,9.28) | 4.01(1.92,6.01) | / | / | / |
| | tiny-2ag-hard | **12.03**(11.37,12.56) | 8.30(6.14,10.11) | 9.43(9.04,9.93) | 4.28(1.74,7.05) | / | / | / |
| | tiny-4ag | **29.95**(29.10,30.85) | 22.93(22.71,23.16) | 16.15(13.47,18.29) | 9.93(7.53,11.76) | / | / | / |
| | tiny-4ag-hard | **20.65**(18.83,21.87) | 13.10(6.99,18.96)* | 14.15(13.60,14.72) | 5.94(2.44,9.61) | / | / | / |
| | small-4ag | 9.90(6.22,13.14)* | **11.70**(11.44,11.98) | 6.18(5.57,6.63) | 2.57(0.94,4.29) | / | / | / |
| | small-4ag-hard | **7.10**(6.28,7.79) | 5.77(4.28,6.88)* | 4.68(4.48,4.90) | 1.27(0.35,2.22) | / | / | / |
| | medium-4ag | **7.71**(6.45,8.60) | 3.74(2.23,5.28) | 4.04(2.71,5.13) | 1.27(0.73,1.75) | / | / | / |
| | medium-4ag-hard | **3.49**(2.69,4.18) | 2.11(1.22,2.97)* | 1.03(0.31,1.84) | 1.41(0.56,2.26) | / | / | / |
| | large-4ag | **3.90**(2.75,4.80) | 3.48(3.10,3.80)* | 1.26(0.66,1.84) | 1.19(0.46,1.93) | / | / | / |
| | large-4ag-hard | **1.84**(1.12,2.49) | 1.20(0.58,1.83)* | 0.00(0.00,0.00) | 0.01(0.00,0.01) | / | / | / |
| | xlarge-4ag | 2.03(1.11,2.90)* | **2.85**(2.51,3.13) | 1.34(0.87,1.76) | 0.00(0.00,0.01) | / | / | / |
| | xlarge-4ag-hard | **0.40**(0.01,0.99) | 0.16(0.00,0.46)* | 0.00(0.00,0.00) | 0.00(0.00,0.01) | / | / | / |
| | medium-6ag | **8.72**(7.45,9.69) | 8.65(7.77,9.34)* | 6.39(6.02,6.74) | 2.29(1.05,3.50) | / | / | / |
| | large-8ag | 7.97(7.79,8.16) | **10.13**(9.75,10.45) | 5.08(4.63,5.54) | 2.97(1.72,4.16) | / | / | / |
| | large-8ag-hard | **5.87**(5.54,6.18) | 4.75(4.20,5.24) | 1.42(0.56,2.35) | 1.59(0.81,2.35) | / | / | / |
| **MaBrax** | hopper_3x1 | 1421.60(1406.21,1439.54) | 1394.73(1341.52,1442.51)* | 1463.84(1375.77,1552.98)* | 1358.66(1321.78,1398.55) | 1553.56(1365.02,1712.54)* | **1556.21**(1509.13,1608.98) | / |
| | halfcheetah_6x1 | 2092.89(1938.95,2223.71) | 1899.52(1635.11,2133.49) | 2335.33(2225.48,2456.97) | 2272.93(2123.10,2402.57) | 2833.44(2663.34,3017.65)* | **3229.46**(2988.55,3484.62) | / |
| | walker2d_2x3 | 663.16(588.64,741.08) | 763.42(672.33,861.29) | 1330.90(1186.47,1467.49)* | 623.44(569.29,671.57) | **1448.05**(1323.51,1584.77) | 1200.39(1104.48,1289.15) | / |
| | ant_4x2 | 2004.15(1826.76,2192.74) | 1564.84(1397.74,1672.30) | 2138.03(2016.37,2258.42) | 2998.59(2824.27,3163.46) | 3553.26(3204.93,3899.81)* | **3964.94**(3641.01,4260.67) | / |
| | humanoid_9—8 | 2066.41(2010.90,2117.48) | 390.82(385.95,395.77) | 463.74(462.19,465.39) | 453.42(447.78,459.02) | **4029.01**(3763.57,4206.90) | 3095.01(2899.63,3294.38) | / |
| **Smax** | 2s3z | **1.96**(1.96,1.96) | 1.64(1.63,1.66) | 1.93(1.92,1.93) | 1.78(1.74,1.81) | / | / | 1.80(1.78,1.81) |
| | 3s5z | **1.91**(1.91,1.91) | 1.69(1.66,1.72) | 1.84(1.84,1.85) | 1.81(1.80,1.83) | / | / | 1.68(1.67,1.69) |
| | 3s_vs_5z | **1.85**(1.85,1.86) | 1.64(1.61,1.67) | 1.66(1.65,1.67) | 1.51(1.48,1.55) | / | / | 1.68(1.66,1.70) |
| | 6h_vs_8z | 1.92(1.92,1.93)* | **1.93**(1.93,1.94) | 1.74(1.73,1.76) | 1.70(1.68,1.73) | / | / | 1.53(1.31,1.69) |
| | 5m_vs_6m | 1.18(0.95,1.42) | 1.17(0.99,1.36) | 0.81(0.65,0.99) | **1.58**(1.50,1.65) | / | / | 1.35(1.22,1.48) |
| | 10m_vs_11m | 1.61(1.44,1.76)* | 1.30(1.26,1.35) | 1.16(1.11,1.20) | **1.63**(1.59,1.67) | / | / | 1.39(1.33,1.43) |
| | 3s5z_vs_3s6z | **1.62**(1.59,1.65) | 1.56(1.49,1.61)* | 1.38(1.35,1.42) | 1.38(1.32,1.44) | / | / | 1.42(1.38,1.46) |
| | 27m_vs_30m | **1.93**(1.91,1.95) | 1.61(1.56,1.66) | 1.63(1.58,1.67) | 1.71(1.62,1.79) | / | / | 1.28(1.17,1.40) |
| | smacv2_5_units | **1.62**(1.61,1.63) | 1.61(1.60,1.62)* | 1.54(1.53,1.55) | 1.55(1.54,1.56) | / | / | 1.50(1.49,1.51) |
| | smacv2_10_units | 1.33(1.29,1.36) | 1.42(1.42,1.43) | **1.48**(1.48,1.49) | 1.33(1.31,1.34) | / | / | 1.30(1.28,1.32) |
| | smacv2_20_units | 1.11(1.05,1.16) | **1.23**(1.22,1.24)* | 1.22(1.21,1.24)* | 0.87(0.85,0.88) | / | / | 0.85(0.80,0.91) |
| **Connector** | con-5x5x3a | **0.85**(0.85,0.85) | 0.67(0.66,0.67) | 0.81(0.81,0.82) | 0.81(0.81,0.82) | / | / | / |
| | con-7x7x5a | **0.79**(0.79,0.79) | 0.66(0.66,0.66) | 0.73(0.73,0.73) | 0.74(0.74,0.74) | / | / | / |
| | con-10x10x10a | **0.71**(0.71,0.71) | 0.18(0.15,0.19) | 0.40(0.39,0.41) | 0.49(0.49,0.49) | / | / | / |
| | con-15x15x23a | **0.64**(0.64,0.64) | -0.13(-0.16,-0.10) | 0.16(0.14,0.18) | 0.24(0.23,0.25) | / | / | / |
| **LBF** | 8x8-2p-2f-coop | **1.00**(1.00,1.00) | 0.95(0.94,0.95) | 0.97(0.97,0.97) | 0.97(0.96,0.97) | / | / | / |
| | 2s-8x8-2p-2f-coop | **1.00**(0.99,1.00) | 0.93(0.93,0.94) | 0.97(0.96,0.97) | 0.96(0.96,0.97) | / | / | / |
| | 10x10-3p-3f | **0.99**(0.99,0.99) | 0.98(0.98,0.98) | 0.95(0.94,0.95) | 0.95(0.94,0.95) | / | / | / |
| | 2s-10x10-3p-3f | **0.98**(0.98,0.98) | 0.91(0.91,0.91) | 0.94(0.93,0.94) | 0.93(0.93,0.94) | / | / | / |
| | 15x15-3p-5f | **0.86**(0.85,0.87) | 0.73(0.72,0.74) | 0.73(0.70,0.75) | 0.66(0.65,0.68) | / | / | / |
| | 15x15-4p-3f | **0.97**(0.97,0.97) | 0.94(0.94,0.94) | 0.93(0.92,0.93) | 0.92(0.91,0.93) | / | / | / |
| | 15x15-4p-5f | **0.92**(0.92,0.93) | 0.84(0.84,0.85) | 0.84(0.84,0.85) | 0.80(0.79,0.81) | / | / | / |
| **MPE** | simple_spread_3ag | -6.81(-6.90,-6.71) | -9.89(-10.16,-9.66) | -9.43(-9.48,-9.39) | -11.18(-11.52,-10.81) | **-5.86**(-5.94,-5.77) | -6.21(-6.48,-5.98) | / |
| | simple_spread_5ag | **-18.50**(-18.92,-18.04) | -29.75(-30.35,-29.06) | -24.25(-24.34,-24.17) | -24.33(-24.44,-24.23) | -25.59(-27.15,-23.58) | -20.89(-22.54,-19.49) | / |
| | simple_spread_10ag | **-40.06**(-40.22,-39.90) | -57.64(-58.21,-57.05) | -42.82(-43.08,-42.60) | -43.30(-43.46,-43.17) | -54.09(-54.42,-53.75) | -52.08(-52.68,-51.24) | / |

## C.2.2 MAX OVER FULL TIMESERIES

Table 3: Maximum episode return over training with 95% bootstrap confidence intervals for all tasks. Bold values indicate the highest score per task and an asterisk indicates that a score overlaps with the highest score within one confidence interval.

| | Task | Sable (Ours) | MAT | MAPPO | IPPO | MASAC | HASAC | QMIX |
|---|---|---|---|---|---|---|---|---|
| **Rware** | tiny-2ag | $\mathbf{22.11}_{(21.32,22.94)}$ | $17.94_{(17.12,18.85)}$ | $13.49_{(9.14,16.90)}$ | $8.47_{(4.15,12.72)}$ | / | / | / |
| | tiny-2ag-hard | $\mathbf{16.81}_{(16.43,17.24)}$ | $14.14_{(12.37,15.32)}$ | $14.60_{(14.17,15.06)}$ | $6.81_{(3.08,10.68)}$ | / | / | / |
| | tiny-4ag | $\mathbf{46.82}_{(45.40,48.08)}$ | $30.69_{(30.33,31.08)}$ | $30.98_{(28.71,33.13)}$ | $20.60_{(16.25,23.58)}$ | / | / | / |
| | tiny-4ag-hard | $\mathbf{33.89}_{(32.67,34.94)}$ | $22.20_{(13.73,29.64)}$ | $22.17_{(21.36,23.12)}$ | $10.67_{(4.65,16.65)}$ | / | / | / |
| | small-4ag | $17.98_{(11.37,22.72)}^{*}$ | $\mathbf{19.49}_{(19.20,19.77)}$ | $11.99_{(11.58,12.44)}$ | $4.29_{(1.57,7.27)}$ | / | / | / |
| | small-4ag-hard | $\mathbf{13.28}_{(12.48,14.08)}$ | $10.72_{(8.28,12.19)}$ | $10.43_{(10.08,10.76)}$ | $2.59_{(0.80,4.57)}$ | / | / | / |
| | medium-4ag | $\mathbf{13.93}_{(12.79,14.75)}$ | $8.12_{(5.52,10.55)}$ | $8.78_{(6.27,10.66)}$ | $2.81_{(1.80,3.75)}$ | / | / | / |
| | medium-4ag-hard | $\mathbf{7.37}_{(6.43,8.21)}$ | $5.04_{(3.23,6.59)}^{*}$ | $3.17_{(1.25,5.18)}$ | $2.05_{(0.82,3.29)}$ | / | / | / |
| | large-4ag | $\mathbf{6.92}_{(5.67,7.87)}$ | $5.38_{(5.17,5.59)}$ | $3.23_{(1.74,4.62)}$ | $1.96_{(0.78,3.14)}$ | / | / | / |
| | large-4ag-hard | $\mathbf{3.82}_{(2.49,4.90)}$ | $2.50_{(1.24,3.72)}^{*}$ | $0.05_{(0.04,0.07)}$ | $0.10_{(0.05,0.15)}$ | / | / | / |
| | xlarge-4ag | $4.03_{(2.47,5.40)}^{*}$ | $\mathbf{5.18}_{(4.81,5.50)}$ | $3.98_{(3.16,4.67)}$ | $0.04_{(0.02,0.06)}$ | / | / | / |
| | xlarge-4ag-hard | $\mathbf{0.82}_{(0.07,1.94)}$ | $0.46_{(0.04,1.19)}^{*}$ | $0.04_{(0.02,0.05)}$ | $0.02_{(0.01,0.02)}$ | / | / | / |
| | medium-6ag | $14.76_{(13.86,15.45)}^{*}$ | $\mathbf{15.28}_{(14.87,15.69)}$ | $13.64_{(13.43,13.87)}$ | $3.90_{(1.85,5.91)}$ | / | / | / |
| | large-8ag | $12.68_{(12.42,13.01)}$ | $\mathbf{16.40}_{(15.97,16.83)}$ | $9.33_{(8.88,9.80)}$ | $5.42_{(3.47,6.91)}$ | / | / | / |
| | large-8ag-hard | $\mathbf{10.24}_{(9.94,10.51)}$ | $9.84_{(9.43,10.25)}^{*}$ | $3.88_{(1.75,6.11)}$ | $4.10_{(2.32,5.68)}$ | / | / | / |
| **MaBrax** | hopper_3x1 | $2210.18_{(2153.99,2277.06)}$ | $1965.98_{(1885.27,2034.57)}$ | $2043.15_{(1835.76,2244.53)}$ | $1684.45_{(1603.88,1766.18)}$ | $2250.96_{(1922.39,2489.34)}^{*}$ | $\mathbf{2423.96}_{(2379.64,2465.84)}$ | / |
| | halfcheetah_6x1 | $2768.53_{(2652.18,2878.64)}$ | $2718.06_{(2527.29,2917.67)}$ | $2916.22_{(2761.25,3090.70)}$ | $2790.42_{(2588.79,2972.60)}$ | $3313.08_{(3065.48,3575.75)}^{*}$ | $\mathbf{3725.47}_{(3381.70,4059.25)}$ | / |
| | walker2d_2x3 | $1078.64_{(903.36,1255.63)}$ | $1301.76_{(1095.03,1495.67)}$ | $\mathbf{2658.69}_{(2289.04,3013.49)}$ | $1093.07_{(1071.24,1123.98)}$ | $2642.20_{(2447.12,2830.37)}^{*}$ | $2238.50_{(2007.02,2493.15)}^{*}$ | / |
| | ant_4x2 | $3697.15_{(3298.29,4093.04)}$ | $2822.87_{(2591.03,3027.88)}$ | $3846.86_{(3622.78,4060.76)}$ | $5200.05_{(4946.75,5447.66)}$ | $5454.61_{(4933.16,5977.32)}^{*}$ | $\mathbf{5797.08}_{(5484.28,6112.50)}$ | / |
| | humanoid_9—8 | $3795.88_{(3656.46,3900.63)}$ | $434.35_{(427.70,442.07)}$ | $551.26_{(546.73,556.93)}$ | $556.80_{(545.92,567.29)}$ | $\mathbf{6257.01}_{(5880.06,6514.80)}$ | $4950.08_{(4544.66,5402.46)}$ | / |
| **Smax** | 2s3z | $\mathbf{2.00}_{(2.00,2.00)}$ | $\mathbf{2.00}_{(2.00,2.00)}$ | $\mathbf{2.00}_{(2.00,2.00)}$ | $\mathbf{2.00}_{(2.00,2.00)}$ | / | / | $\mathbf{2.00}_{(2.00,2.00)}$ |
| | 3s5z | $\mathbf{2.00}_{(2.00,2.00)}$ | $\mathbf{2.00}_{(2.00,2.00)}$ | $\mathbf{2.00}_{(2.00,2.00)}$ | $\mathbf{2.00}_{(2.00,2.00)}$ | / | / | $\mathbf{2.00}_{(2.00,2.00)}$ |
| | 3s_vs_5z | $\mathbf{2.00}_{(2.00,2.00)}$ | $\mathbf{2.00}_{(2.00,2.00)}$ | $\mathbf{2.00}_{(2.00,2.00)}$ | $\mathbf{2.00}_{(2.00,2.00)}$ | / | / | $\mathbf{2.00}_{(2.00,2.00)}$ |
| | 6h_vs_8z | $\mathbf{2.00}_{(2.00,2.00)}$ | $\mathbf{2.00}_{(2.00,2.00)}$ | $\mathbf{2.00}_{(2.00,2.00)}$ | $\mathbf{2.00}_{(2.00,2.00)}$ | / | / | $1.87_{(1.63,2.00)}^{*}$ |
| | 5m_vs_6m | $1.61_{(1.31,1.90)}$ | $1.72_{(1.45,1.97)}$ | $1.18_{(0.87,1.51)}$ | $\mathbf{2.00}_{(1.99,2.00)}$ | / | / | $1.87_{(1.78,1.96)}$ |
| | 10m_vs_11m | $1.88_{(1.75,2.00)}^{*}$ | $1.92_{(1.81,2.00)}^{*}$ | $1.49_{(1.43,1.55)}$ | $1.98_{(1.97,2.00)}^{*}$ | / | / | $\mathbf{2.00}_{(2.00,2.00)}$ |
| | 3s5z_vs_3s6z | $\mathbf{2.00}_{(2.00,2.00)}$ | $1.99_{(1.98,2.00)}^{*}$ | $1.99_{(1.98,2.00)}^{*}$ | $1.98_{(1.97,1.99)}$ | / | / | $1.91_{(1.87,1.94)}$ |
| | 27m_vs_30m | $\mathbf{2.00}_{(2.00,2.00)}$ | $1.96_{(1.92,1.99)}$ | $1.95_{(1.91,1.99)}$ | $1.98_{(1.95,2.00)}^{*}$ | / | / | $1.82_{(1.76,1.89)}$ |
| | smacv2_5_units | $\mathbf{1.96}_{(1.94,1.97)}$ | $1.92_{(1.90,1.94)}^{*}$ | $1.93_{(1.91,1.95)}^{*}$ | $1.90_{(1.89,1.90)}$ | / | / | $1.88_{(1.86,1.90)}$ |
| | smacv2_10_units | $1.79_{(1.78,1.81)}$ | $1.80_{(1.78,1.82)}$ | $\mathbf{1.90}_{(1.88,1.92)}$ | $1.81_{(1.78,1.83)}$ | / | / | $1.79_{(1.77,1.82)}$ |
| | smacv2_20_units | $\mathbf{1.70}_{(1.64,1.77)}$ | $1.69_{(1.65,1.72)}^{*}$ | $1.69_{(1.65,1.72)}^{*}$ | $1.29_{(1.27,1.32)}$ | / | / | $1.28_{(1.15,1.40)}$ |
| **Connector** | con-5x5x3a | $\mathbf{0.89}_{(0.89,0.90)}$ | $0.88_{(0.87,0.88)}^{*}$ | $\mathbf{0.89}_{(0.88,0.89)}$ | $\mathbf{0.89}_{(0.88,0.89)}$ | / | / | / |
| | con-7x7x5a | $\mathbf{0.85}_{(0.85,0.85)}$ | $0.82_{(0.81,0.84)}$ | $0.83_{(0.82,0.84)}$ | $0.83_{(0.83,0.84)}$ | / | / | / |
| | con-10x10x10a | $\mathbf{0.79}_{(0.78,0.80)}$ | $0.34_{(0.32,0.37)}$ | $0.58_{(0.57,0.59)}$ | $0.64_{(0.63,0.65)}$ | / | / | / |
| | con-15x15x23a | $\mathbf{0.74}_{(0.74,0.75)}$ | $0.02_{(0.00,0.03)}$ | $0.34_{(0.32,0.35)}$ | $0.43_{(0.42,0.44)}$ | / | / | / |
| **LBF** | 8x8-2p-2f-coop | $\mathbf{1.00}_{(1.00,1.00)}$ | $\mathbf{1.00}_{(1.00,1.00)}$ | $\mathbf{1.00}_{(1.00,1.00)}$ | $\mathbf{1.00}_{(1.00,1.00)}$ | / | / | / |
| | 2s-8x8-2p-2f-coop | $\mathbf{1.00}_{(1.00,1.00)}$ | $\mathbf{1.00}_{(1.00,1.00)}$ | $\mathbf{1.00}_{(1.00,1.00)}$ | $\mathbf{1.00}_{(1.00,1.00)}$ | / | / | / |
| | 10x10-3p-3f | $\mathbf{1.00}_{(1.00,1.00)}$ | $\mathbf{1.00}_{(1.00,1.00)}$ | $\mathbf{1.00}_{(1.00,1.00)}$ | $\mathbf{1.00}_{(1.00,1.00)}$ | / | / | / |
| | 2s-10x10-3p-3f | $\mathbf{1.00}_{(1.00,1.00)}$ | $\mathbf{1.00}_{(1.00,1.00)}$ | $\mathbf{1.00}_{(1.00,1.00)}$ | $\mathbf{1.00}_{(1.00,1.00)}$ | / | / | / |
| | 15x15-3p-5f | $\mathbf{1.00}_{(1.00,1.00)}$ | $0.98_{(0.97,0.99)}$ | $0.99_{(0.98,1.00)}^{*}$ | $0.96_{(0.94,0.97)}$ | / | / | / |
| | 15x15-4p-3f | $\mathbf{1.00}_{(1.00,1.00)}$ | $\mathbf{1.00}_{(1.00,1.00)}$ | $\mathbf{1.00}_{(1.00,1.00)}$ | $\mathbf{1.00}_{(1.00,1.00)}$ | / | / | / |
| | 15x15-4p-5f | $\mathbf{1.00}_{(1.00,1.00)}$ | $\mathbf{1.00}_{(1.00,1.00)}$ | $\mathbf{1.00}_{(1.00,1.00)}$ | $\mathbf{1.00}_{(1.00,1.00)}$ | / | / | / |
| **MPE** | simple_spread_3ag | $\mathbf{-4.32}_{(-4.46,-4.16)}$ | $-5.85_{(-5.98,-5.73)}$ | $-5.96_{(-6.07,-5.84)}$ | $-7.35_{(-7.71,-6.92)}$ | $-4.61_{(-4.68,-4.54)}$ | $-4.56_{(-4.68,-4.46)}^{*}$ | / |
| | simple_spread_5ag | $\mathbf{-11.97}_{(-12.50,-11.43)}$ | $-23.97_{(-25.04,-22.70)}$ | $-20.98_{(-21.22,-20.70)}$ | $-20.54_{(-20.71,-20.37)}$ | $-18.74_{(-21.05,-16.15)}$ | $-16.53_{(-18.18,-15.19)}$ | / |
| | simple_spread_10ag | $\mathbf{-35.32}_{(-35.63,-35.02)}$ | $-48.12_{(-49.24,-47.17)}$ | $-38.94_{(-39.28,-38.55)}$ | $-39.55_{(-39.82,-39.28)}$ | $-49.39_{(-49.80,-48.93)}$ | $-47.76_{(-48.61,-46.40)}$ | / |

### C.2.3 FINAL VALUE OF THE TIMESERIES

Table 4: Final episode return over training with 95% bootstrap confidence intervals for all tasks. Bold values indicate the highest score per task and an asterisk indicates that a score overlaps with the highest score within one confidence interval.

| | Task | Sable (Ours) | MAT | MAPPO | IPPO | MASAC | HASAC | QMIX |
|---|---|---|---|---|---|---|---|---|
| **Rware** | tiny-2ag | $20.92_{(19.85,22.02)}$ | $17.26_{(16.10,18.48)}$ | $12.01_{(8.12,15.07)}$ | $7.65_{(3.76,11.51)}$ | / | / | / |
| | tiny-2ag-hard | $16.14_{(15.80,16.46)}$ | $13.52_{(11.68,14.66)}$ | $13.77_{(13.32,14.18)}$ | $6.07_{(2.66,9.66)}$ | / | / | / |
| | tiny-4ag | $43.65_{(42.24,45.10)}$ | $28.31_{(27.52,29.07)}$ | $26.60_{(24.54,28.54)}$ | $17.70_{(13.62,21.07)}$ | / | / | / |
| | tiny-4ag-hard | $30.63_{(28.88,32.44)}$ | $20.63_{(12.79,27.37)}$ | $19.11_{(18.34,19.88)}$ | $9.32_{(3.98,14.58)}$ | / | / | / |
| | small-4ag | $17.08_{(10.82,21.56)}^{*}$ | $18.60_{(17.99,19.12)}$ | $10.72_{(9.87,11.61)}$ | $3.82_{(1.35,6.42)}$ | / | / | / |
| | small-4ag-hard | $12.47_{(11.73,13.20)}$ | $9.35_{(7.17,10.71)}$ | $9.13_{(8.85,9.39)}$ | $2.28_{(0.71,4.03)}$ | / | / | / |
| | medium-4ag | $13.05_{(12.08,13.79)}$ | $7.38_{(5.09,9.45)}$ | $7.40_{(5.32,9.00)}$ | $2.59_{(1.59,3.51)}$ | / | / | / |
| | medium-4ag-hard | $6.80_{(5.93,7.61)}$ | $4.69_{(3.00,6.16)}^{*}$ | $2.88_{(1.13,4.67)}$ | $1.86_{(0.73,2.98)}$ | / | / | / |
| | large-4ag | $6.30_{(5.07,7.25)}$ | $4.49_{(4.16,4.81)}$ | $3.07_{(1.61,4.43)}$ | $1.82_{(0.71,2.97)}$ | / | / | / |
| | large-4ag-hard | $3.48_{(2.24,4.53)}$ | $2.37_{(1.14,3.54)}^{*}$ | $0.00_{(0.00,0.00)}$ | $0.04_{(0.01,0.08)}$ | / | / | / |
| | xlarge-4ag | $3.77_{(2.32,5.03)}^{*}$ | $4.71_{(4.44,4.96)}$ | $3.67_{(2.82,4.37)}$ | $0.01_{(0.00,0.02)}$ | / | / | / |
| | xlarge-4ag-hard | $0.74_{(0.02,1.81)}$ | $0.37_{(0.01,1.02)}^{*}$ | $0.00_{(0.00,0.01)}$ | $0.00_{(0.00,0.00)}$ | / | / | / |
| | medium-6ag | $12.43_{(11.33,13.28)}^{*}$ | $13.12_{(12.52,13.68)}$ | $12.34_{(11.73,12.99)}^{*}$ | $3.62_{(1.71,5.49)}$ | / | / | / |
| | large-8ag | $10.62_{(9.77,11.47)}$ | $15.08_{(14.58,15.65)}$ | $8.42_{(7.80,9.05)}$ | $4.97_{(3.17,6.35)}$ | / | / | / |
| | large-8ag-hard | $9.52_{(9.27,9.78)}$ | $9.21_{(8.74,9.66)}^{*}$ | $3.58_{(1.60,5.62)}$ | $3.43_{(1.93,4.79)}$ | / | / | / |
| **MaBrax** | hopper_3x1 | $2090.31_{(2032.93,2146.79)}$ | $1891.11_{(1806.49,1971.11)}$ | $1869.63_{(1648.22,2081.73)}^{*}$ | $1562.81_{(1495.04,1631.14)}$ | $1460.16_{(970.98,1939.99)}$ | $1493.91_{(1132.95,1824.75)}$ | / |
| | halfcheetah_6x1 | $2388.03_{(1885.96,2751.80)}$ | $2552.63_{(2139.07,2884.05)}$ | $2914.90_{(2762.87,3084.95)}$ | $2779.89_{(2580.41,2957.09)}$ | $3272.96_{(3019.66,3540.14)}^{*}$ | $3633.09_{(3277.97,3977.98)}$ | / |
| | walker2d_2x3 | $1026.17_{(866.81,1193.51)}$ | $1128.18_{(953.49,1289.64)}$ | $2506.84_{(2141.45,2856.99)}$ | $407.46_{(368.42,451.87)}$ | $1564.60_{(1261.78,1895.57)}$ | $1285.74_{(938.79,1605.32)}$ | / |
| | ant_4x2 | $3006.37_{(2411.98,3579.62)}$ | $2299.52_{(1882.49,2637.99)}$ | $3346.00_{(2995.72,3682.63)}$ | $4397.34_{(3823.41,4901.51)}^{*}$ | $4821.93_{(4277.66,5350.59)}$ | $4627.60_{(4270.06,4979.85)}^{*}$ | / |
| | humanoid_9—8 | $3368.27_{(3139.82,3557.73)}$ | $385.01_{(371.68,399.06)}$ | $520.27_{(507.95,535.05)}$ | $521.47_{(500.32,540.73)}$ | $5046.19_{(4316.12,5668.46)}$ | $3928.96_{(3210.23,4559.54)}^{*}$ | / |
| **Smax** | 2s3z | $2.00_{(2.00,2.00)}$ | $1.99_{(1.98,2.00)}^{*}$ | $1.99_{(1.98,2.00)}^{*}$ | $1.99_{(1.98,2.00)}^{*}$ | / | / | $2.00_{(1.99,2.00)}$ |
| | 3s5z | $1.99_{(1.97,2.00)}^{*}$ | $2.00_{(1.99,2.00)}$ | $2.00_{(2.00,2.00)}$ | $2.00_{(2.00,2.00)}$ | / | / | $1.99_{(1.98,2.00)}^{*}$ |
| | 3s_vs_5z | $1.98_{(1.97,1.99)}^{*}$ | $1.95_{(1.93,1.97)}^{*}$ | $1.99_{(1.97,2.00)}$ | $1.99_{(1.98,2.00)}$ | / | / | $1.94_{(1.92,1.97)}^{*}$ |
| | 6h_vs_8z | $2.00_{(2.00,2.00)}$ | $1.99_{(1.98,2.00)}^{*}$ | $2.00_{(1.99,2.00)}$ | $1.99_{(1.98,2.00)}^{*}$ | / | / | $1.59_{(1.28,1.85)}$ |
| | 5m_vs_6m | $1.25_{(0.90,1.62)}$ | $1.61_{(1.29,1.90)}^{*}$ | $1.07_{(0.78,1.41)}$ | $1.89_{(1.86,1.93)}$ | / | / | $1.74_{(1.62,1.86)}^{*}$ |
| | 10m_vs_11m | $1.84_{(1.68,1.98)}^{*}$ | $1.87_{(1.74,1.96)}^{*}$ | $1.29_{(1.21,1.37)}$ | $1.73_{(1.64,1.82)}$ | / | / | $1.93_{(1.90,1.96)}$ |
| | 3s5z_vs_3s6z | $1.94_{(1.91,1.97)}$ | $1.92_{(1.88,1.95)}^{*}$ | $1.92_{(1.86,1.97)}^{*}$ | $1.91_{(1.86,1.95)}^{*}$ | / | / | $1.68_{(1.60,1.75)}$ |
| | 27m_vs_30m | $2.00_{(2.00,2.00)}$ | $1.91_{(1.87,1.95)}$ | $1.86_{(1.81,1.91)}$ | $1.91_{(1.83,1.97)}$ | / | / | $1.42_{(1.22,1.63)}$ |
| | smacv2_5_units | $1.80_{(1.78,1.83)}$ | $1.79_{(1.75,1.83)}^{*}$ | $1.70_{(1.66,1.74)}$ | $1.73_{(1.69,1.77)}$ | / | / | $1.70_{(1.66,1.74)}$ |
| | smacv2_10_units | $1.59_{(1.55,1.65)}$ | $1.64_{(1.56,1.70)}$ | $1.77_{(1.74,1.80)}$ | $1.61_{(1.54,1.67)}$ | / | / | $1.58_{(1.51,1.65)}$ |
| | smacv2_20_units | $1.48_{(1.41,1.56)}^{*}$ | $1.49_{(1.43,1.56)}^{*}$ | $1.53_{(1.46,1.59)}$ | $1.11_{(1.05,1.18)}$ | / | / | $1.01_{(0.87,1.15)}$ |
| **Connector** | con-5x5x3a | $0.86_{(0.84,0.87)}$ | $0.81_{(0.78,0.84)}^{*}$ | $0.83_{(0.82,0.85)}^{*}$ | $0.84_{(0.83,0.85)}^{*}$ | / | / | / |
| | con-7x7x5a | $0.80_{(0.79,0.82)}$ | $0.75_{(0.73,0.77)}$ | $0.76_{(0.75,0.78)}$ | $0.77_{(0.74,0.79)}$ | / | / | / |
| | con-10x10x10a | $0.73_{(0.72,0.75)}$ | $0.24_{(0.19,0.29)}$ | $0.47_{(0.44,0.50)}$ | $0.56_{(0.55,0.57)}$ | / | / | / |
| | con-15x15x23a | $0.70_{(0.68,0.72)}$ | $-0.14_{(-0.21,-0.08)}$ | $0.21_{(0.16,0.26)}$ | $0.36_{(0.32,0.39)}$ | / | / | / |
| **LBF** | 8x8-2p-2f-coop | $1.00_{(1.00,1.00)}$ | $1.00_{(1.00,1.00)}$ | $1.00_{(1.00,1.00)}$ | $1.00_{(1.00,1.00)}$ | / | / | / |
| | 2s-8x8-2p-2f-coop | $1.00_{(1.00,1.00)}$ | $0.99_{(0.99,1.00)}^{*}$ | $1.00_{(1.00,1.00)}$ | $1.00_{(1.00,1.00)}$ | / | / | / |
| | 10x10-3p-3f | $1.00_{(1.00,1.00)}$ | $1.00_{(1.00,1.00)}$ | $1.00_{(1.00,1.00)}$ | $1.00_{(1.00,1.00)}$ | / | / | / |
| | 2s-10x10-3p-3f | $1.00_{(0.99,1.00)}$ | $0.97_{(0.96,0.98)}$ | $1.00_{(1.00,1.00)}$ | $0.99_{(0.99,1.00)}^{*}$ | / | / | / |
| | 15x15-3p-5f | $0.97_{(0.96,0.99)}$ | $0.93_{(0.91,0.95)}$ | $0.96_{(0.95,0.98)}^{*}$ | $0.92_{(0.89,0.94)}$ | / | / | / |
| | 15x15-4p-3f | $1.00_{(1.00,1.00)}$ | $1.00_{(0.99,1.00)}$ | $0.99_{(0.98,1.00)}^{*}$ | $0.99_{(0.98,1.00)}^{*}$ | / | / | / |
| | 15x15-4p-5f | $0.99_{(0.99,1.00)}$ | $0.97_{(0.95,0.99)}^{*}$ | $0.97_{(0.95,0.99)}^{*}$ | $0.97_{(0.95,0.98)}$ | / | / | / |
| **MPE** | simple_spread_3ag | $-5.05_{(-5.32,-4.76)}$ | $-6.78_{(-7.06,-6.48)}$ | $-6.85_{(-7.22,-6.51)}$ | $-8.49_{(-9.19,-7.77)}$ | $-5.13_{(-5.26,-4.96)}^{*}$ | $-6.06_{(-7.26,-5.15)}^{*}$ | / |
| | simple_spread_5ag | $-12.61_{(-13.11,-12.15)}$ | $-25.37_{(-26.60,-23.99)}$ | $-22.56_{(-23.05,-22.04)}$ | $-21.79_{(-22.27,-21.30)}$ | $-19.99_{(-22.78,-17.03)}$ | $-17.94_{(-19.39,-16.69)}$ | / |
| | simple_spread_10ag | $-36.75_{(-37.17,-36.32)}$ | $-49.44_{(-50.46,-48.36)}$ | $-41.06_{(-42.03,-39.98)}$ | $-41.40_{(-42.22,-40.53)}$ | $-52.86_{(-54.17,-51.69)}$ | $-50.17_{(-51.32,-48.47)}$ | / |

## C.2.4 ABSOLUTE METRIC

Table 5: Absolute episode return over training with 95% bootstrap confidence intervals for all tasks. Bold values indicate the highest score per task and an asterisk indicates that a score overlaps with the highest score within one confidence interval.

| | Task | Sable (Ours) | MAT | MAPPO | IPPO | MASAC | HASAC | QMIX |
|---|---|---|---|---|---|---|---|---|
| **Rware** | tiny-2ag | $\mathbf{21.17}_{(20.42,21.95)}$ | $17.06_{(16.10,18.09)}$ | $12.28_{(8.20,15.48)}$ | $7.61_{(3.68,11.46)}$ | / | / | / |
| | tiny-2ag-hard | $\mathbf{15.93}_{(15.50,16.41)}$ | $13.44_{(11.74,14.58)}$ | $13.60_{(13.19,14.08)}$ | $6.15_{(2.76,9.74)}$ | / | / | / |
| | tiny-4ag | $\mathbf{43.56}_{(41.80,45.10)}$ | $28.19_{(27.57,28.82)}$ | $26.29_{(24.38,27.92)}$ | $16.98_{(13.42,19.44)}$ | / | / | / |
| | tiny-4ag-hard | $\mathbf{30.97}_{(29.91,31.96)}$ | $20.54_{(12.70,27.44)}$ | $19.01_{(18.23,19.85)}$ | $9.06_{(3.98,14.05)}$ | / | / | / |
| | small-4ag | $16.47_{(10.45,20.80)}{}^{*}$ | $\mathbf{18.27}_{(17.92,18.57)}$ | $10.52_{(10.10,11.03)}$ | $3.69_{(1.32,6.27)}$ | / | / | / |
| | small-4ag-hard | $\mathbf{12.02}_{(11.28,12.78)}$ | $9.68_{(7.46,10.98)}$ | $9.44_{(9.23,9.66)}$ | $2.27_{(0.69,4.03)}$ | / | / | / |
| | medium-4ag | $\mathbf{12.74}_{(11.72,13.41)}$ | $7.62_{(5.17,9.91)}$ | $7.82_{(5.60,9.49)}$ | $2.58_{(1.68,3.41)}$ | / | / | / |
| | medium-4ag-hard | $\mathbf{6.79}_{(5.89,7.54)}$ | $4.64_{(2.96,6.09)}{}^{*}$ | $2.80_{(1.13,4.55)}$ | $1.89_{(0.75,3.05)}$ | / | / | / |
| | large-4ag | $\mathbf{6.22}_{(5.03,7.14)}$ | $4.61_{(4.46,4.78)}$ | $3.02_{(1.58,4.39)}$ | $1.84_{(0.73,2.96)}$ | / | / | / |
| | large-4ag-hard | $\mathbf{3.46}_{(2.22,4.46)}$ | $2.28_{(1.09,3.40)}{}^{*}$ | $0.00_{(0.00,0.01)}$ | $0.05_{(0.01,0.09)}$ | / | / | / |
| | xlarge-4ag | $3.76_{(2.27,5.09)}{}^{*}$ | $\mathbf{4.71}_{(4.42,4.96)}$ | $3.73_{(2.94,4.40)}$ | $0.01_{(0.00,0.02)}$ | / | / | / |
| | xlarge-4ag-hard | $\mathbf{0.70}_{(0.01,1.74)}$ | $0.39_{(0.01,1.07)}{}^{*}$ | $0.00_{(0.00,0.00)}$ | $0.00_{(0.00,0.00)}$ | / | / | / |
| | medium-6ag | $12.97_{(12.26,13.52)}{}^{*}$ | $\mathbf{13.32}_{(12.93,13.70)}$ | $12.13_{(11.82,12.48)}$ | $3.47_{(1.65,5.24)}$ | / | / | / |
| | large-8ag | $11.01_{(10.70,11.33)}$ | $\mathbf{14.72}_{(14.27,15.24)}$ | $8.35_{(7.95,8.77)}$ | $4.87_{(3.10,6.20)}$ | / | / | / |
| | large-8ag-hard | $\mathbf{9.22}_{(8.93,9.52)}$ | $9.07_{(8.61,9.49)}{}^{*}$ | $3.38_{(1.51,5.35)}$ | $3.63_{(2.04,5.02)}$ | / | / | / |
| **MaBrax** | hopper_3x1 | $2053.29_{(2012.38,2099.07)}$ | $1901.02_{(1822.85,1963.89)}$ | $1933.73_{(1752.82,2108.94)}$ | $1608.52_{(1545.00,1673.47)}$ | $2253.33_{(1924.08,2492.43)}{}^{*}$ | $\mathbf{2459.92}_{(2400.82,2520.81)}$ | / |
| | halfcheetah_6x1 | $2717.51_{(2592.24,2830.35)}$ | $2709.90_{(2515.66,2911.42)}$ | $2912.64_{(2758.24,3086.38)}$ | $2784.17_{(2585.20,2963.31)}$ | $3311.72_{(3068.37,3567.81)}{}^{*}$ | $\mathbf{3739.89}_{(3398.86,4071.03)}$ | / |
| | walker2d_2x3 | $1051.15_{(889.00,1216.13)}$ | $1177.26_{(998.46,1353.89)}$ | $2483.05_{(2117.17,2838.47)}{}^{*}$ | $1086.30_{(1064.04,1117.18)}$ | $\mathbf{2636.95}_{(2421.05,2867.14)}$ | $2310.41_{(2075.84,2555.89)}{}^{*}$ | / |
| | ant_4x2 | $3185.75_{(2794.94,3599.22)}$ | $2428.97_{(2197.26,2615.22)}$ | $3307.23_{(3105.04,3504.37)}$ | $4366.81_{(4080.06,4636.39)}$ | $4751.03_{(4281.89,5223.50)}{}^{*}$ | $\mathbf{5069.32}_{(4679.84,5442.11)}$ | / |
| | humanoid_9—8 | $3449.78_{(3279.82,3618.87)}$ | $397.25_{(392.76,401.83)}$ | $515.45_{(509.01,520.98)}$ | $512.65_{(500.90,524.68)}$ | $\mathbf{6302.42}_{(5818.37,6646.02)}$ | $4983.49_{(4625.08,5392.85)}$ | / |
| **Smax** | 2s3z | $\mathbf{2.00}_{(2.00,2.00)}$ | $1.99_{(1.99,2.00)}{}^{*}$ | $\mathbf{2.00}_{(1.99,2.00)}$ | $\mathbf{2.00}_{(2.00,2.00)}$ | / | / | $\mathbf{2.00}_{(1.99,2.00)}$ |
| | 3s5z | $\mathbf{2.00}_{(1.99,2.00)}$ | $\mathbf{2.00}_{(2.00,2.00)}$ | $\mathbf{2.00}_{(2.00,2.00)}$ | $\mathbf{2.00}_{(2.00,2.00)}$ | / | / | $\mathbf{2.00}_{(2.00,2.00)}$ |
| | 3s_vs_5z | $1.98_{(1.98,1.99)}{}^{*}$ | $1.96_{(1.95,1.97)}$ | $1.98_{(1.98,1.99)}{}^{*}$ | $\mathbf{1.99}_{(1.98,1.99)}$ | / | / | $1.95_{(1.93,1.97)}$ |
| | 6h_vs_8z | $\mathbf{2.00}_{(2.00,2.00)}$ | $1.99_{(1.98,1.99)}{}^{*}$ | $\mathbf{2.00}_{(1.99,2.00)}$ | $1.99_{(1.99,2.00)}{}^{*}$ | / | / | $1.85_{(1.64,1.96)}$ |
| | 5m_vs_6m | $1.47_{(1.13,1.80)}$ | $1.60_{(1.29,1.88)}$ | $1.05_{(0.75,1.39)}$ | $\mathbf{1.91}_{(1.90,1.92)}$ | / | / | $1.75_{(1.62,1.87)}$ |
| | 10m_vs_11m | $1.80_{(1.62,1.98)}{}^{*}$ | $1.86_{(1.70,1.97)}{}^{*}$ | $1.28_{(1.22,1.32)}$ | $1.89_{(1.87,1.91)}$ | / | / | $\mathbf{1.96}_{(1.93,1.98)}$ |
| | 3s5z_vs_3s6z | $\mathbf{1.94}_{(1.93,1.95)}$ | $1.93_{(1.91,1.95)}{}^{*}$ | $1.90_{(1.86,1.93)}$ | $1.89_{(1.87,1.92)}$ | / | / | $1.79_{(1.75,1.82)}$ |
| | 27m_vs_30m | $\mathbf{2.00}_{(1.99,2.00)}$ | $1.87_{(1.80,1.92)}$ | $1.81_{(1.74,1.87)}$ | $1.91_{(1.84,1.98)}$ | / | / | $1.70_{(1.63,1.78)}$ |
| | smacv2_5_units | $1.72_{(1.70,1.74)}$ | $\mathbf{1.77}_{(1.75,1.78)}$ | $1.70_{(1.68,1.71)}$ | $1.69_{(1.68,1.71)}$ | / | / | $1.68_{(1.65,1.70)}$ |
| | smacv2_10_units | $1.52_{(1.48,1.56)}$ | $1.62_{(1.59,1.65)}$ | $\mathbf{1.70}_{(1.68,1.71)}$ | $1.54_{(1.53,1.56)}$ | / | / | $1.60_{(1.56,1.63)}$ |
| | smacv2_20_units | $1.47_{(1.40,1.52)}{}^{*}$ | $\mathbf{1.49}_{(1.46,1.53)}$ | $1.42_{(1.38,1.45)}$ | $1.07_{(1.03,1.10)}$ | / | / | $1.11_{(0.99,1.23)}$ |
| **Connector** | con-5x5x3a | $\mathbf{0.85}_{(0.85,0.86)}$ | $0.81_{(0.80,0.82)}$ | $0.83_{(0.82,0.84)}$ | $0.83_{(0.83,0.84)}$ | / | / | / |
| | con-7x7x5a | $\mathbf{0.79}_{(0.79,0.80)}$ | $0.75_{(0.74,0.76)}$ | $0.75_{(0.74,0.76)}$ | $0.76_{(0.75,0.77)}$ | / | / | / |
| | con-10x10x10a | $\mathbf{0.74}_{(0.74,0.74)}$ | $0.65_{(0.63,0.67)}$ | $0.70_{(0.70,0.70)}$ | $0.71_{(0.71,0.72)}$ | / | / | / |
| | con-15x15x23a | $\mathbf{0.70}_{(0.70,0.71)}$ | $0.25_{(0.18,0.31)}$ | $0.63_{(0.62,0.64)}$ | $0.67_{(0.67,0.67)}$ | / | / | / |
| **LBF** | 8x8-2p-2f-coop | $\mathbf{1.00}_{(1.00,1.00)}$ | $\mathbf{1.00}_{(1.00,1.00)}$ | $\mathbf{1.00}_{(1.00,1.00)}$ | $\mathbf{1.00}_{(1.00,1.00)}$ | / | / | / |
| | 2s-8x8-2p-2f-coop | $\mathbf{1.00}_{(1.00,1.00)}$ | $1.00_{(0.99,1.00)}$ | $\mathbf{1.00}_{(1.00,1.00)}$ | $\mathbf{1.00}_{(1.00,1.00)}$ | / | / | / |
| | 10x10-3p-3f | $\mathbf{1.00}_{(1.00,1.00)}$ | $0.99_{(0.99,1.00)}{}^{*}$ | $\mathbf{1.00}_{(1.00,1.00)}$ | $\mathbf{1.00}_{(1.00,1.00)}$ | / | / | / |
| | 2s-10x10-3p-3f | $0.99_{(0.99,1.00)}{}^{*}$ | $0.97_{(0.96,0.97)}$ | $\mathbf{1.00}_{(1.00,1.00)}$ | $0.99_{(0.99,0.99)}$ | / | / | / |
| | 15x15-3p-5f | $0.96_{(0.96,0.97)}{}^{*}$ | $0.91_{(0.90,0.92)}$ | $\mathbf{0.97}_{(0.95,0.97)}$ | $0.90_{(0.88,0.92)}$ | / | / | / |
| | 15x15-4p-3f | $\mathbf{1.00}_{(1.00,1.00)}$ | $0.99_{(0.99,1.00)}{}^{*}$ | $0.99_{(0.99,1.00)}{}^{*}$ | $\mathbf{1.00}_{(0.99,1.00)}$ | / | / | / |
| | 15x15-4p-5f | $\mathbf{0.99}_{(0.99,0.99)}$ | $0.97_{(0.96,0.97)}$ | $0.98_{(0.97,0.98)}$ | $0.97_{(0.97,0.97)}$ | / | / | / |
| **MPE** | simple_spread_3ag | $\mathbf{-4.92}_{(-5.11,-4.74)}$ | $-6.59_{(-6.74,-6.46)}$ | $-6.72_{(-6.86,-6.59)}$ | $-8.35_{(-8.84,-7.81)}$ | $-5.27_{(-5.34,-5.20)}$ | $-5.29_{(-5.44,-5.14)}$ | / |
| | simple_spread_5ag | $\mathbf{-12.75}_{(-13.32,-12.20)}$ | $-25.30_{(-26.32,-24.19)}$ | $-22.84_{(-22.98,-22.70)}$ | $-21.97_{(-22.27,-21.68)}$ | $-19.89_{(-22.41,-17.11)}$ | $-17.85_{(-19.71,-16.43)}$ | / |
| | simple_spread_10ag | $\mathbf{-36.93}_{(-37.13,-36.73)}$ | $-50.07_{(-51.20,-49.10)}$ | $-41.83_{(-42.15,-41.52)}$ | $-42.08_{(-42.37,-41.82)}$ | $-51.01_{(-51.52,-50.54)}$ | $-49.71_{(-50.72,-48.23)}$ | / |

## C.2.5 INTER-QUARTILE MEAN OVER TIMESERIES

Table 6: Inter-quartile mean episode return over training with 95% bootstrap confidence intervals for all tasks. Bold values indicate the highest score per task and an asterisk indicates that a score overlaps with the highest score within one confidence interval.

| | Task | Sable (Ours) | MAT | MAPPO | IPPO | MASAC | HASAC | QMIX |
|---|---|---|---|---|---|---|---|---|
| Rware | tiny-2ag | **$17.68_{(17.17,18.07)}$** | $12.67_{(10.68,14.29)}$ | $7.80_{(5.01,10.43)}$ | $4.19_{(2.01,6.30)}$ | / | / | / |
| | tiny-2ag-hard | **$13.67_{(12.92,14.25)}$** | $9.40_{(6.53,11.75)}$ | $11.40_{(10.89,12.06)}$ | $4.99_{(1.97,8.29)}$ | / | / | / |
| | tiny-4ag | **$32.81_{(31.85,33.78)}$** | $25.69_{(25.51,25.87)}$ | $17.36_{(13.72,20.10)}$ | $10.47_{(7.72,12.55)}$ | / | / | / |
| | tiny-4ag-hard | **$23.12_{(20.86,24.59)}$** | $14.17_{(7.18,21.05)}{}^{*}$ | $15.71_{(15.08,16.35)}$ | $6.53_{(2.40,10.77)}$ | / | / | / |
| | small-4ag | $10.75_{(6.44,14.54)}{}^{*}$ | **$13.39_{(13.02,13.78)}$** | $6.75_{(5.82,7.40)}$ | $2.83_{(1.07,4.71)}$ | / | / | / |
| | small-4ag-hard | **$7.64_{(6.67,8.44)}$** | $6.40_{(4.64,7.80)}{}^{*}$ | $4.81_{(4.50,5.13)}$ | $1.39_{(0.35,2.43)}$ | / | / | / |
| | medium-4ag | **$8.34_{(6.75,9.50)}$** | $3.82_{(2.11,5.59)}$ | $4.27_{(2.78,5.57)}$ | $1.28_{(0.70,1.83)}$ | / | / | / |
| | medium-4ag-hard | **$3.69_{(2.69,4.51)}$** | $2.05_{(1.06,3.07)}{}^{*}$ | $0.89_{(0.21,1.69)}$ | $1.68_{(0.66,2.71)}{}^{*}$ | / | / | / |
| | large-4ag | **$4.28_{(2.82,5.39)}$** | $3.91_{(3.49,4.25)}{}^{*}$ | $1.16_{(0.57,1.77)}$ | $1.42_{(0.55,2.32)}$ | / | / | / |
| | large-4ag-hard | **$1.96_{(1.10,2.79)}$** | $1.27_{(0.55,1.96)}{}^{*}$ | $0.00_{(0.00,0.00)}$ | $0.00_{(0.00,0.00)}$ | / | / | / |
| | xlarge-4ag | $2.21_{(1.14,3.22)}{}^{*}$ | **$3.24_{(2.77,3.63)}$** | $1.06_{(0.58,1.57)}$ | $0.00_{(0.00,0.00)}$ | / | / | / |
| | xlarge-4ag-hard | **$0.44_{(0.00,1.08)}$** | $0.14_{(0.00,0.41)}{}^{*}$ | $0.00_{(0.00,0.00)}{}^{*}$ | $0.00_{(0.00,0.00)}{}^{*}$ | / | / | / |
| | medium-6ag | **$9.58_{(7.95,10.82)}$** | $9.44_{(8.27,10.35)}{}^{*}$ | $6.72_{(6.21,7.19)}$ | $2.47_{(1.10,3.86)}$ | / | / | / |
| | large-8ag | $9.06_{(8.87,9.25)}$ | **$11.26_{(10.89,11.58)}$** | $5.84_{(5.24,6.43)}$ | $3.18_{(1.63,4.73)}$ | / | / | / |
| | large-8ag-hard | **$6.76_{(6.36,7.12)}$** | $5.11_{(4.19,5.91)}$ | $1.27_{(0.47,2.21)}$ | $1.52_{(0.65,2.40)}$ | / | / | / |
| MaBrax | hopper_3x1 | $1469.95_{(1455.05,1487.71)}{}^{*}$ | $1433.88_{(1363.94,1495.89)}{}^{*}$ | $1536.67_{(1441.48,1636.33)}{}^{*}$ | $1437.85_{(1391.41,1486.83)}{}^{*}$ | **$1621.99_{(1419.72,1802.02)}$** | $1613.30_{(1550.07,1682.11)}{}^{*}$ | / |
| | halfcheetah_6x1 | $2216.42_{(2059.24,2355.54)}$ | $2034.59_{(1753.67,2284.10)}$ | $2485.63_{(2357.15,2622.30)}$ | $2493.34_{(2311.06,2654.75)}$ | $2930.42_{(2752.13,3123.16)}{}^{*}$ | **$3376.90_{(3107.51,3660.88)}$** | / |
| | walker2d_2x3 | $668.68_{(593.98,748.42)}$ | $766.61_{(672.31,875.02)}$ | $1294.99_{(1165.42,1420.19)}{}^{*}$ | $571.58_{(509.62,629.28)}$ | **$1470.89_{(1345.06,1611.62)}$** | $1229.26_{(1128.54,1322.44)}$ | / |
| | ant_4x2 | $2087.42_{(1907.36,2285.86)}$ | $1585.95_{(1401.20,1705.86)}$ | $2196.26_{(2048.08,2340.83)}$ | $3210.00_{(3012.26,3390.20)}$ | $3828.43_{(3404.10,4259.06)}{}^{*}$ | **$4150.12_{(3787.47,4476.47)}$** | / |
| | humanoid_9—8 | $2200.64_{(2142.08,2260.57)}$ | $390.75_{(385.77,395.82)}$ | $471.11_{(469.67,472.49)}$ | $458.76_{(452.74,465.10)}$ | **$4181.90_{(3909.18,4369.39)}$** | $3255.90_{(3045.68,3467.54)}$ | / |
| Smax | 2s3z | **$2.00_{(2.00,2.00)}$** | $1.90_{(1.87,1.91)}$ | **$2.00_{(2.00,2.00)}$** | $1.99_{(1.98,1.99)}$ | / | / | $1.99_{(1.98,1.99)}$ |
| | 3s5z | **$2.00_{(2.00,2.00)}$** | $1.93_{(1.90,1.95)}$ | $1.99_{(1.99,2.00)}{}^{*}$ | **$2.00_{(2.00,2.00)}$** | / | / | $1.98_{(1.98,1.99)}$ |
| | 3s_vs_5z | **$1.98_{(1.97,1.98)}$** | $1.87_{(1.84,1.90)}$ | $1.92_{(1.91,1.93)}$ | $1.75_{(1.70,1.80)}$ | / | / | $1.90_{(1.89,1.91)}$ |
| | 6h_vs_8z | **$2.00_{(2.00,2.00)}$** | $1.99_{(1.99,1.99)}$ | $1.99_{(1.98,1.99)}$ | $1.94_{(1.92,1.96)}$ | / | / | $1.62_{(1.38,1.80)}$ |
| | 5m_vs_6m | $1.26_{(0.97,1.56)}$ | $1.19_{(0.95,1.45)}$ | $0.87_{(0.68,1.09)}$ | **$1.75_{(1.64,1.83)}$** | / | / | $1.59_{(1.40,1.77)}{}^{*}$ |
| | 10m_vs_11m | **$1.73_{(1.53,1.92)}$** | $1.31_{(1.26,1.35)}$ | $1.19_{(1.13,1.23)}$ | $1.72_{(1.66,1.77)}{}^{*}$ | / | / | $1.59_{(1.49,1.67)}{}^{*}$ |
| | 3s5z_vs_3s6z | **$1.81_{(1.77,1.84)}$** | $1.76_{(1.66,1.83)}{}^{*}$ | $1.52_{(1.47,1.57)}$ | $1.52_{(1.44,1.61)}$ | / | / | $1.62_{(1.57,1.66)}$ |
| | 27m_vs_30m | **$1.99_{(1.97,2.00)}$** | $1.75_{(1.69,1.82)}$ | $1.73_{(1.67,1.78)}$ | $1.83_{(1.72,1.92)}$ | / | / | $1.42_{(1.26,1.58)}$ |
| | smacv2_5_units | **$1.70_{(1.69,1.71)}$** | $1.67_{(1.66,1.68)}$ | $1.63_{(1.63,1.64)}$ | $1.63_{(1.63,1.64)}$ | / | / | $1.61_{(1.60,1.61)}$ |
| | smacv2_10_units | $1.40_{(1.35,1.44)}$ | $1.49_{(1.48,1.50)}$ | **$1.59_{(1.58,1.60)}$** | $1.38_{(1.36,1.39)}$ | / | / | $1.41_{(1.38,1.43)}$ |
| | smacv2_20_units | $1.11_{(1.05,1.19)}$ | **$1.32_{(1.30,1.33)}$** | $1.29_{(1.28,1.30)}{}^{*}$ | $0.89_{(0.87,0.91)}$ | / | / | $0.87_{(0.82,0.92)}$ |
| Connector | con-5x5x3a | **$0.85_{(0.85,0.85)}$** | $0.75_{(0.74,0.76)}$ | $0.83_{(0.83,0.83)}$ | $0.83_{(0.83,0.83)}$ | / | / | / |
| | con-7x7x5a | **$0.79_{(0.79,0.80)}$** | $0.72_{(0.71,0.72)}$ | $0.75_{(0.75,0.75)}$ | $0.76_{(0.76,0.76)}$ | / | / | / |
| | con-10x10x10a | **$0.73_{(0.73,0.73)}$** | $0.19_{(0.17,0.21)}$ | $0.43_{(0.42,0.44)}$ | $0.52_{(0.52,0.53)}$ | / | / | / |
| | con-15x15x23a | **$0.69_{(0.68,0.69)}$** | $-0.11_{(-0.14,-0.08)}$ | $0.18_{(0.17,0.20)}$ | $0.26_{(0.24,0.27)}$ | / | / | / |
| LBF | 8x8-2p-2f-coop | **$1.00_{(1.00,1.00)}$** | **$1.00_{(1.00,1.00)}$** | **$1.00_{(1.00,1.00)}$** | **$1.00_{(1.00,1.00)}$** | / | / | / |
| | 2s-8x8-2p-2f-coop | **$1.00_{(1.00,1.00)}$** | $0.99_{(0.99,0.99)}$ | **$1.00_{(1.00,1.00)}$** | **$1.00_{(1.00,1.00)}$** | / | / | / |
| | 10x10-3p-3f | **$1.00_{(1.00,1.00)}$** | **$1.00_{(1.00,1.00)}$** | **$1.00_{(1.00,1.00)}$** | **$1.00_{(1.00,1.00)}$** | / | / | / |
| | 2s-10x10-3p-3f | **$0.99_{(0.99,1.00)}$** | $0.95_{(0.95,0.95)}$ | $0.98_{(0.94,0.99)}{}^{*}$ | $0.98_{(0.97,0.98)}$ | / | / | / |
| | 15x15-3p-5f | **$0.92_{(0.91,0.93)}$** | $0.77_{(0.76,0.79)}$ | $0.80_{(0.77,0.83)}$ | $0.73_{(0.71,0.75)}$ | / | / | / |
| | 15x15-4p-3f | **$1.00_{(1.00,1.00)}$** | **$1.00_{(1.00,1.00)}$** | **$1.00_{(1.00,1.00)}$** | **$1.00_{(0.99,1.00)}$** | / | / | / |
| | 15x15-4p-5f | **$0.98_{(0.97,0.98)}$** | $0.91_{(0.91,0.92)}$ | $0.88_{(0.87,0.89)}$ | $0.91_{(0.90,0.92)}$ | / | / | / |
| MPE | simple_spread_3ag | $-5.60_{(-5.69,-5.50)}$ | $-8.22_{(-8.57,-7.95)}$ | $-8.16_{(-8.22,-8.10)}$ | $-10.02_{(-10.43,-9.57)}$ | **$-5.36_{(-5.42,-5.30)}$** | $-5.71_{(-5.96,-5.49)}$ | / |
| | simple_spread_5ag | **$-17.15_{(-17.65,-16.62)}$** | $-29.51_{(-30.20,-28.72)}$ | $-23.65_{(-23.73,-23.57)}$ | $-23.47_{(-23.59,-23.36)}$ | $-24.98_{(-26.88,-22.52)}$ | $-18.96_{(-20.85,-17.33)}{}^{*}$ | / |
| | simple_spread_10ag | **$-38.81_{(-38.94,-38.68)}$** | $-57.15_{(-57.80,-56.47)}$ | $-42.25_{(-42.52,-42.02)}$ | $-42.52_{(-42.69,-42.37)}$ | $-53.31_{(-53.61,-53.01)}$ | $-51.13_{(-51.74,-50.23)}$ | / |

# D HYPERPARAMETERS

We make all hyperparameters as well as instructions for rerunning all benchmarks available along with the code provided at the following link: https://sites.google.com/view/sable-marl. For all on-policy algorithms on all tasks, we always use 128 effective vectorised environments. For HASAC and MASAC we use 64 vectorised environments while for QMIX we use 32 vectorised environments. We leverage the design architecture of Mava which can distribute the end-to-end RL training loop over multiple devices using the pmap JAX transformation and also vectorise it using the vmap JAX transformation. For IPPO, MAPPO and Sable we train systems with and without memory. For IPPO and MAPPO this means that networks include a Gated Recurrent Unit (GRU) (Cho, 2014) layer for memory and for Sable this means training over full episode trajectories at a time or only one timestep at a time. For MASAC and HASAC (Liu et al., 2023a) we only train policies using MLPs and for QMIX (Rashid et al., 2020a) we only train a system with memory due to the original implementations of these algorithms doing so.

In cases where systems are trained with and without memory, we report results for the version of the system that performs the best on a given task. In all hyperparameter tables a parameter marked with an asterisk "*" implies that it is only relevant for the memory version of a given algorithm.

## D.1 HYPERPARAMETER OPTIMISATION

We use the same default parameters and parameter search spaces for a given algorithm on all tasks. All algorithms are tuned for 40 trials on each task using the Tree-structured Parzen Estimator (TPE) Bayesian optimisation algorithm from the Optuna library (Akiba et al., 2019).

### D.1.1 DEFAULT PARAMETERS

For all algorithms we use the default parameters:

Table 7: Default hyperparameters for Sable.

| Parameter | Value |
| --- | --- |
| Activation function | GeLU |
| Normalise Aavantage | True |
| Value function coefficient | 0.5 |
| Discount $\gamma$ | 0.99 |
| GAE $\lambda$ | 0.9 |
| Rollout length | 128 |
| Add one-hot agent ID | True |

Table 8: Default hyperparameters for MAT.

| Parameter | Value |
| --- | --- |
| Activation function | GeLU |
| Normalise advantage | True |
| Value function coefficient | 0.5 |
| Discount $\gamma$ | 0.99 |
| GAE $\lambda$ | 0.9 |
| Rollout length | 128 |
| Add one-hot agent ID | True |

Table 9: Default hyperparameters for MAPPO and IPPO.

| Parameter | Value |
| --- | --- |
| Critic network layer sizes | [128, 128] |
| Policy network layer sizes | [128, 128] |
| Number of recurrent layers* | 1 |
| Size of recurrent layer* | 128 |
| Activation Function | ReLU |
| Normalise advantage | True |
| Value function coefficient | 0.5 |
| Discount $\gamma$ | 0.99 |
| GAE $\lambda$ | 0.9 |
| Rollout length | 128 |
| Add one-hot agent ID | True |

Table 10: Default hyperparameters for MASAC and HASAC.

| Parameter | Value |
| --- | --- |
| Q-network layer sizes | [128, 128] |
| Policy network layer sizes | [128, 128] |
| Activation function | ReLU |
| Replay buffer size | 100000 |
| Rollout length | 8 |
| Maximum gradient norm | 10 |
| Add One-hot Agent ID | True |

Table 11: Default hyperparameters for QMIX.

| Parameter | Value |
|---|---|
| Q-network layer sizes | [128, 128] |
| Number of recurrent layers* | 1 |
| Size of recurrent layer* | 256 |
| Activation function | ReLU |
| Maximum gradient norm | 10 |
| Add one-hot agent ID | True |
| Sample sequence length | 20 |
| Hard target update | False |
| Polyak averaging coefficient $\tau$ | 0.01 |
| Minimum exploration value $\epsilon$ | 0.05 |
| Exploration value decay rate | 0.00001 |
| Rollout length | 2 |
| Epochs | 2 |
| Add one-hot agent ID | True |

### D.1.2 SEARCH SPACES

We always use discrete search spaces and search over the following parameters per algorithm

Table 12: Hyperparameter Search Space for Sable.

| Parameter | Value |
|---|---|
| PPO epochs | {2, 5, 10, 15} |
| Number of minibatches | {1, 2, 4, 8} |
| Entropy coefficient | {0.1, 0.01, 0.001, 1} |
| Clipping $\epsilon$ | {0.05, 0.1, 0.2} |
| Maximum gradient norm | {0.5, 5, 10} |
| Learning rate | {1e-3, 5e-4, 2.5e-4, 1e-4, 1e-5} |
| Model embedding dimension | {32, 64, 128} |
| Number retention heads | {1, 2, 4} |
| Number retention blocks | {1, 2, 3} |
| Retention heads $\kappa$ scaling parameter | {0.3, 0.5, 0.8, 1} |

Table 13: Hyperparameter Search Space for MAT.

| Parameter | Value |
|---|---|
| PPO epochs | {2, 5, 10, 15} |
| Number of minibatches | {1, 2, 4, 8} |
| Entropy coefficient | {0.1, 0.01, 0.001, 1} |
| Clipping $\epsilon$ | {0.05, 0.1, 0.2} |
| Maximum gradient norm | {0.5, 5, 10} |
| Learning rate | {1e-3, 5e-4, 2.5e-4, 1e-4, 1e-5} |
| Model embedding dimension | {32, 64, 128} |
| Number transformer heads | {1, 2, 4} |
| Number transformer blocks | {1, 2, 3} |

Table 14: Hyperparameter Search Space for MAPPO and IPPO.

| Parameter | Value |
|-----------|-------|
| PPO epochs | {2, 4, 8} |
| Number of minibatches | {2, 4, 8} |
| Entropy coefficient | {0, 0.01, 0.00001} |
| Clipping $\epsilon$ | {0.05, 0.1, 0.2} |
| Maximum gradient norm | {0.5, 5, 10} |
| Critic learning rate | {1e-4, 2.5e-4, 5e-4} |
| Policy learning rate | {1e-4, 2.5e-4, 5e-4} |
| Recurrent chunk size | {8, 16, 32, 64, 128} |

Table 15: Hyperparameter Search Space for MASAC and HASAC.

| Parameter | Value |
|-----------|-------|
| Epochs | {32, 64, 128} |
| Batch size | {32, 64, 128} |
| Policy update delay | {1, 2, 4} |
| Policy learning rate | {1e-3, 3e-4, 5e-4} |
| Q-network learning rate | {1e-3, 3e-4, 5e-4} |
| Alpha learning rate | {1e-3, 3e-4, 5e-4} |
| Polyak averaging coefficient $\tau$ | {0.001, 0.005} |
| Discount factor $\gamma$ | {0.99, 0.95} |
| Autotune alpha | {True, False} |
| Target entropy scale | {1, 2, 5, 10} |
| Initial alpha | {0.0005, 0.005, 0.1} |
| Shuffle agents (HASAC only) | {True, False} |

Table 16: Hyperparameter Search Space for QMIX.

| Parameter | Value |
|-----------|-------|
| Batch size | {16, 32, 64, 128} |
| Q-network learning rate | {3e-3, 3e-4, 3e-5, 3e-6} |
| Replay buffer size | {2000, 4000, 8000} |
| Target network update period | {100, 200, 400, 800} |
| Mixer network embedding dimension | {32, 64} |

## D.2 COMPUTATIONAL RESOURCES

Experiments were run using various machines that either had NVIDIA Quadro RTX 4000 (8GB), Tesla V100 (32GB) or A100 (80GB) GPUs as well on TPU v4-8 and v3-8 devices.

# E SABLE IMPLEMENTATION DETAILS

## E.1 PSEUDOCODE

A useful note when reading this pseudocode is that bold inputs represent that the item is *joint*, in other words, it applies to all agents. For example: $\boldsymbol{a}$ is the joint action and $\boldsymbol{v}$ is the value of all agents.

The clipped PPO policy objective can be given as:

$$L_p(\theta, \hat{A}, \boldsymbol{o}_t, \boldsymbol{a}_t) = \min\left(r_t(\theta)\hat{A}_t, \text{clip}(r_t(\theta), 1 \pm \epsilon)\hat{A}_t\right)$$

$$\text{where} \quad r_t(\theta, \boldsymbol{o}_t, \boldsymbol{a}_t) = \frac{\pi_\theta\left(\boldsymbol{a}_t | \boldsymbol{o}_t\right)}{\pi_{\theta_{old}}\left(\boldsymbol{a}_t | \boldsymbol{o}_t\right)} \tag{7}$$

The encoder is optimised using the mean squared error:

$$L_v(\phi, \boldsymbol{v}, \hat{\boldsymbol{v}}) = \left(\boldsymbol{v}_\phi(\boldsymbol{o}_t) - \hat{\boldsymbol{v}}_t\right)^2 \tag{8}$$

where $\hat{\boldsymbol{v}}_t$ is the value target computed as $\hat{\boldsymbol{v}}_t = r_t + \gamma \boldsymbol{v}(\boldsymbol{o}_{t+1})$. We always compute the advantage estimate and value targets using generalised advantage estimation (GAE) (Schulman et al., 2015). We denote $d_t$ as a binary flag indicating whether the current episode has ended or not, with $d_t = 1$ signifying episode termination and $d_t = 0$ indicating continuation.

---

**Algorithm 1** Sable

**Require:** rollout length ($L$) updates ($U$) agents ($N$) epochs ($K$) minibatches ($M$)
1: $h_{joint}^{act} \leftarrow h_{enc}^{act}, h_{dec}^{act} \leftarrow 0$        ▷ *Initialize hidden state for encoder and decoder to zeros*
2: **for** Update $= 1, 2, \ldots, U$ **do**
3:     $h_{joint}^{train} \leftarrow h_{joint}^{act}$        ▷ *Store initial hidden states for training*
4:     **for** $t = 1, 2, \ldots, L$ **do**        ▷ *Performed in parallel with multiple environments*
5:        $\hat{\boldsymbol{o}}_t, \boldsymbol{v}_t, h_{enc}^{act} \leftarrow \text{encoder.chunkwise}(\boldsymbol{o}_t, h_{enc}^{act})$
6:        **for** $i = 1, 2, \ldots, N$ **do**        ▷ *Auto-regressively decode each agent's action*
7:           $a_t^i, \pi_{old}^i(a_t^i|\hat{o}_t^i), h_{dec}^{act} \leftarrow \text{decoder.recurrent}(\hat{o}_t^i, a_t^{i-1}, h_{dec}^{act})$
8:        Step environment using joint action $\boldsymbol{a}_t$ to produce $\boldsymbol{o}_{t+1}, r_t, d_t$
9:        Store $(\boldsymbol{o}_t, \boldsymbol{a}_t, d_t, r_t, \boldsymbol{\pi}_{old}(\boldsymbol{a_t}|\hat{\boldsymbol{o}}_t))$ in buffer $\mathcal{B}$
10:        **if** episode terminates **then** $h_{joint} \leftarrow 0$ **else** $h_{joint} \leftarrow \kappa h_{joint}$
11:     Use $GAE$ to compute advantage estimates $\hat{A}$ and value targets $\hat{\boldsymbol{v}}$
12:     **for** $1, 2, \ldots, K$ **do**
13:        Sample trajectories $\tau = (\boldsymbol{o}_{b_{1:L}}, \boldsymbol{a}_{b_{1:L}}, d_{b_{1:L}}, r_{b_{1:L}}, \boldsymbol{\pi}_{old}(\boldsymbol{a}_{b_{1:L}}|\hat{\boldsymbol{o}}_{b_{1:L}}))$ from $\mathcal{B}$
14:        Within each trajectory $\tau$ shuffle all items along the agent dimension
15:        **for** $1, 2, \ldots, M$ **do**
16:           Generate $D_{enc}, D_{dec}$ given Equations 9 and 10, and $d_{b_{1:L}}$
17:           $\hat{\boldsymbol{o}}_{1:L}, \boldsymbol{v}_{b_{1:L}} \leftarrow \text{encoder.chunkwise}(\boldsymbol{o}_{b_{1:L}}, h_{enc}^{train}, D_{enc})$
18:           $\boldsymbol{\pi}(\boldsymbol{a}_{b_{1:L}}|\hat{\boldsymbol{o}}_{b_{1:L}}) \leftarrow \text{decoder.chunkwise}(\boldsymbol{a}_{b_{1:L}}, \hat{\boldsymbol{o}}_{1:L}, h_{dec}^{train}, D_{dec})$
19:           $\theta \leftarrow \theta + \nabla_\theta L_p(\theta, \hat{A}_{1:L}, \hat{\boldsymbol{o}}_{1:L}, \boldsymbol{a}_{1:L})$
20:           $\phi \leftarrow \phi + \nabla_\phi L_v(\phi, r_{b_{1:L}}, \boldsymbol{v}_{b_{1:L}}, \hat{\boldsymbol{v}}_{b_{1:L}})$

---

### E.2 Additional Implementation Insights

#### E.2.1 Retentive Encoder-Decoder Architecture

RetNet is a decoder-only architecture designed with only causal language modelling in mind. This is illustrated by the assumption in Equations 1, 2 and 3 that the $key$, $query$ and $value$ inputs are identical, as they are all represented by the single value $x$. However, MAT uses an encoder-decoder model to encode observations and decode actions. In order to extend retention to support the cross-retention used decoder, Sable takes a key, query and value as input to a retention block in place of $X$. Additionally, it uses the key as a proxy for $X$ in the swish gate used in multi-scale retention (Sun et al., 2023) and the query as a proxy for $X$ for the skip connection in the RetNet block (Sun et al., 2023).

#### E.2.2 Adapting the decay matrix for MARL

Since the decay matrix represents the importance of past observations, it is critical to construct it correctly during training, taking into account multiple agents and episode terminations, so that no agent is "favoured" and memory doesn't flow over episode boundaries. To achieve this, we make three modifications to the original decay matrix formula. First, in cooperative MARL, a joint action is formed such that all agents act simultaneously from the perspective of the environment. This means the memory of past observations within the same timestep should be weighted equally between agents; therefore, Sable uses equal decay values for these observations. Second, unlike self-supervised learning, RL requires algorithms to handle episode termination. During acting this is trivial for Sable: hidden states must be reset to zero on the first step of every episode as seen in Equations 4 and 5. However, during training, resetting needs to be performed over the full trajectory $\tau$ in parallel using the decay matrix. If there is a termination on timestep $t_d$, then the decay matrix should be reset from index $(Nt_d, Nt_d)$. Combining these two modifications, we obtain the following equation, given a set of terminal timesteps $\mathbf{T_d}$, then $\forall t_d \in \mathbf{T_d}$:

$$D_{ij} = M_{ij} \odot \tilde{D}_{ij}, \quad M_{ij} = \begin{cases} 0 & \text{if } i \geq Nt_d > j \\ 1 & \text{otherwise} \end{cases}, \quad \tilde{D}_{ij} = \begin{cases} \kappa^{\lfloor (i-j)/N \rfloor}, & \text{if } i \geq j \\ 0, & \text{if } i < j \end{cases} \quad (9)$$

This updated equation makes sure that Sable does not prioritise certain agent's past observations because of their arbitrary ordering and ensures that all observations before the terminal timestep are forgotten.

The final decay matrix modification is only required in the encoder to allow for full self-retention over all agents' observations in a single timestep, to match the full self-attention used in MAT's encoder. Since RetNet is a decoder only model where the decay matrix acts as a causal mask, we had to adjust Sable's architecture to be able to perform self-retention. We do this by creating $N \times N$ blocks within the decay matrix, where each block represents a timestep of $N$ agents. This leads to the final modification of the decay matrix specifically for the encoder:

$$D_{ij} = M_{ij} \odot \hat{D}_{ij}, \quad \hat{D}_{ij} = \begin{cases} \kappa^{\lfloor (i-j)/N \rfloor}, & \text{if } \lfloor i/N \rfloor \geq \lfloor j/N \rfloor \\ 0, & \text{otherwise} \end{cases} \quad (10)$$

In this case, the floor operator in the first condition creates the blocks that enable full self-retention. Examples of both the encoder and decoder decay matrices used during training can be found in appendix E.3.3.

#### E.2.3 Positional Encoding

Positional encodings are crucial for Sable to obtain a notion of time. Previous works in single-agent reinforcement learning (RL) that utilize transformers or similar architectures (Parisotto et al. (2020); Lu et al. (2024)) have demonstrated the importance of positional encoding. Empirical results showed that the best method for sable is absolute positional encoding, as introduced by Vaswani et al. (2017). In this method, the agent's timestep is encoded and added to the key, query, and value during processing. Importantly, we discovered that providing all agents within the same timestep with the same positional encoding is pivotal for performance. When agents were assigned sequential indices (e.g., agent $i$ at step $t$ receiving index $N \times t + i$), as is commonly done with tokens in NLP tasks, the performance was significantly worse.

### E.3 ALGORITHMIC WALKTHROUGH: CONCRETE EXAMPLE

Sable processes full episodes as sequences, chunking them into segments of defined rollout lengths. During each training phase, it processes a chunk and retains hidden states, which are passed forward to subsequent updates. In this section, we provide a concrete example to illustrate how the algorithm works in practice.

#### E.3.1 EXAMPLE SETUP

To illustrate Sable's chunkwise processing, consider a simple environment with 3 agents, and rollout length ($L$) of 4 timesteps. The goal of this setup is to demonstrate how Sable processes the execution and training phases of a trajectory $\tau$, starting at timestep $l$. For this example, we will assume that an episode termination happened at $l + 1$, which means at the second step of the trajectory.

#### E.3.2 EXECUTION PHASE

During this phase, Sable interacts with the environment to build the trajectory $\tau$. Each timestep $t \in \{l, ..., l + 3\}$ is processed sequentially. At each timestep $t$, the encoder takes as input the observation of all the agents at $t$, along with the hidden state from previous timestep, $h_{t-1}^{enc}$. If $t = l$, indicating the beginning of the current execution phase, the encoder uses $h_{l-1}^{enc}$. The encoder then computes the current observation representations $\hat{o}_t$, observation values $v_t$, and updates the hidden state for the next timestep, $h_t^{enc}$. These observation representations are passed to the decoder alongside $h_{t-1}^{dec}$. If $t = l$, the decoder initialises with $h_{l-1}^{dec}$, similar to the encoder. The decoder processes the input recurrently for each agent, generating actions one by one based on the previous agent's action. For the first agent, a start of sequence token is sent to the decoder, signaling that this is the initial agent and prompting the decoder to begin reasoning for its action independently of prior agents. As each agent's action is decoded, an intermediate hidden state $\hat{h}$ is updated, and once the decoder iterates over all agents, it generates $h_t^{dec}$ for the timestep $t$. At the end of each timestep, both the encoder and decoder hidden states are decayed by the factor $\kappa$, reducing the influence of past timesteps. If the episode ends at timestep $t$ (in our case at $t = l + 1$), the hidden states are reset to zero; otherwise, they continue to propagate to the next timestep with the decay applied.

#### E.3.3 TRAINING PHASE

Once the trajectory $\tau$ is collected, the observations from all agents and timesteps are concatenated to form a full trajectory sequence. In this case, the resulting sequence is $[o_l^1, o_l^2, ..., o_{l+3}^2, o_{l+3}^3]$. Both encoder and decoder compute retention over this sequence using Equation 6. However, rather than recalculating or initialising $H_{\tau_{prev}}$, the encoder and decoder take as input the hidden states from the final timestep of the previous execution phase, $h_{l-1}^{enc}$ and $h_{l-1}^{dec}$, respectively. Within the retention mechanism used during training, the decay matrix $D$ and $\xi$ control the decaying and resetting of information.

The decay matrix, $D$, has a shape of $(NL, NL)$, which in this case results in a (12,12) matrix, where each element calculates how much one token retains the information of another token. For example, $D_{5,3}$ shows how much information $o_{l+1}^2$ retains from $o_l^3$:

$$D_{enc} = \begin{pmatrix} \kappa^0 & \kappa^0 & \kappa^0 & 0 & 0 & 0 & 0 & 0 & 0 & 0 & 0 & 0 \\ \kappa^0 & \kappa^0 & \kappa^0 & 0 & 0 & 0 & 0 & 0 & 0 & 0 & 0 & 0 \\ \kappa^0 & \kappa^0 & \kappa^0 & 0 & 0 & 0 & 0 & 0 & 0 & 0 & 0 & 0 \\ \kappa^1 & \kappa^1 & \kappa^1 & \kappa^0 & \kappa^0 & \kappa^0 & 0 & 0 & 0 & 0 & 0 & 0 \\ \kappa^1 & \kappa^1 & \kappa^1 & \kappa^0 & \kappa^0 & \kappa^0 & 0 & 0 & 0 & 0 & 0 & 0 \\ \kappa^1 & \kappa^1 & \kappa^1 & \kappa^0 & \kappa^0 & \kappa^0 & 0 & 0 & 0 & 0 & 0 & 0 \\ 0 & 0 & 0 & 0 & 0 & 0 & \kappa^0 & \kappa^0 & \kappa^0 & 0 & 0 & 0 \\ 0 & 0 & 0 & 0 & 0 & 0 & \kappa^0 & \kappa^0 & \kappa^0 & 0 & 0 & 0 \\ 0 & 0 & 0 & 0 & 0 & 0 & \kappa^0 & \kappa^0 & \kappa^0 & 0 & 0 & 0 \\ 0 & 0 & 0 & 0 & 0 & 0 & \kappa^1 & \kappa^1 & \kappa^1 & \kappa^0 & \kappa^0 & \kappa^0 \\ 0 & 0 & 0 & 0 & 0 & 0 & \kappa^1 & \kappa^1 & \kappa^1 & \kappa^0 & \kappa^0 & \kappa^0 \\ 0 & 0 & 0 & 0 & 0 & 0 & \kappa^1 & \kappa^1 & \kappa^1 & \kappa^0 & \kappa^0 & \kappa^0 \end{pmatrix}$$

$$D_{dec} = \begin{pmatrix} \kappa^0 & 0 & 0 & 0 & 0 & 0 & 0 & 0 & 0 & 0 & 0 & 0 \\ \kappa^0 & \kappa^0 & 0 & 0 & 0 & 0 & 0 & 0 & 0 & 0 & 0 & 0 \\ \kappa^0 & \kappa^0 & \kappa^0 & 0 & 0 & 0 & 0 & 0 & 0 & 0 & 0 & 0 \\ \kappa^1 & \kappa^1 & \kappa^1 & \kappa^0 & 0 & 0 & 0 & 0 & 0 & 0 & 0 & 0 \\ \kappa^1 & \kappa^1 & \kappa^1 & \kappa^0 & \kappa^0 & 0 & 0 & 0 & 0 & 0 & 0 & 0 \\ \kappa^1 & \kappa^1 & \kappa^1 & \kappa^0 & \kappa^0 & \kappa^0 & 0 & 0 & 0 & 0 & 0 & 0 \\ 0 & 0 & 0 & 0 & 0 & 0 & \kappa^0 & 0 & 0 & 0 & 0 & 0 \\ 0 & 0 & 0 & 0 & 0 & 0 & \kappa^0 & \kappa^0 & 0 & 0 & 0 & 0 \\ 0 & 0 & 0 & 0 & 0 & 0 & \kappa^0 & \kappa^0 & \kappa^0 & 0 & 0 & 0 \\ 0 & 0 & 0 & 0 & 0 & 0 & \kappa^1 & \kappa^1 & \kappa^1 & \kappa^0 & 0 & 0 \\ 0 & 0 & 0 & 0 & 0 & 0 & \kappa^1 & \kappa^1 & \kappa^1 & \kappa^0 & \kappa^0 & 0 \\ 0 & 0 & 0 & 0 & 0 & 0 & \kappa^1 & \kappa^1 & \kappa^1 & \kappa^0 & \kappa^0 & \kappa^0 \end{pmatrix}$$

As shown in this example the decay matrix $D$, agents within the same timestep share identical decay values, ensuring consistent retention for all agents within a given timestep. Additionally, the decay matrix resets for agents once an episode ends. For instance, after the termination at timestep $l + 1$, subsequent timesteps $(D_{7:12,1:12})$ no longer retain information from the prior episode as can be seen from the zeros at $D_{7:12,1:6}$. The encoder decay matrix, $D_{enc}$ enables full self-retention over all agents in the same timestep as can be seen through the blocks of equal values within the decay matrix. In contrast, the decoder decay matrix, $D_{dec}$, only allows information to flow backwards in time, so agents can only view the actions of previous agents, as they make decisions sequentially.

The second key variable in the retention mechanism is $\xi$, which controls how much each token in the sequence retains information from the hidden state of the previous chunk ($h_{l-1}^{enc}$ for the encoder and $h_{l-1}^{dec}$ for the decoder). The $\xi$ matrix represents the contribution of past hidden states to the current timestep, ensuring continuity across chunk boundaries. For this case, $\xi$ is structured as follows:

$$\xi = \begin{pmatrix} \kappa^1 \\ \kappa^1 \\ \kappa^1 \\ \kappa^2 \\ \kappa^2 \\ \kappa^2 \\ 0 \\ 0 \\ 0 \\ 0 \\ 0 \\ 0 \end{pmatrix}$$

As shown, $\xi$ ensures that after the termination at timestep $l + 1$, the tokens in subsequent timesteps no longer retain information from the hidden states of the prior episode.

### E.3.4   HANDLING LONG TIMESTEP SEQUENCES

Consider a trajectory batch $\tau'$ consisting of 3 agents and 512 timesteps. This results in a sequence length of $512 \times 3 = 1536$. Given the potential memory limitations of the computational resources, handling such a large sequence may pose challenges when applying Equation 6, which assumes the entire sequence is processed as input.

To handle these long sequences, we divide the trajectory into $i$ smaller chunks. However, it's essential to maintain the condition that each chunk must be organized by timesteps, meaning all agents from the same timestep must belong to the same chunk. The first chunk will use the hidden state as described earlier ($h_{l-1}^{enc}$ for the encoder and $h_{l-1}^{dec}$ for the decoder). For subsequent chunks, when chunking the trajectory into smaller chunks of size 4, the input hidden state is recalculated using the following equation:

$$h_i = K_{[i]}^T (V_{[i]} \odot \zeta) + \delta\kappa^4 h_{i_{prev}}, \quad \zeta = D_{12,1:12}$$

Suppose that a chunk $B$ starts at timestep $b$ and there is a termination at $b + 1$. In this case, the decay matrix for $B$ would be structured similarly to the one used in the example of Section E.3.3. And given that, $\zeta$ will be equal the following :

$$\zeta = \begin{pmatrix} 0 & 0 & 0 & 0 & 0 & 0 & \kappa^1 & \kappa^1 & \kappa^1 & \kappa^0 & \kappa^0 & \kappa^0 \end{pmatrix}$$

This structure ensures that only tokens from the current episode contribute to the hidden state, while any information from the previous episode is ignored. Additionally, the term $\delta\kappa^L H_{B_{prev}}$ is set to zero, ensuring that once an episode ends, the associated hidden state does not carry over to the next chunk.

