# OpenReview forum: "Performant, Memory Efficient and Scalable Multi-Agent Reinforcement Learning"
_ICLR.cc/2025/Conference — Submitted to ICLR 2025_

### Official Review · Reviewer_7eua · 2024-10-23

**Soundness:** 3
**Presentation:** 3
**Contribution:** 2
**Rating:** 6
**Confidence:** 3

**Summary:**

This paper introduces a novel and theoretically sound algorithm Sable, adapting the retention mechanism from retentive networks to MARL. The authors demonstrate that Sable ranks first in 34 out of 45 tasks (roughly 75%), outperforming SOTA MARL algorithms across 6 environments: RWARE, LBF, MABrax, SMAX, Connector and MPE. Meanwhile, the advancements of Sable in memory efficiency and agent scalability are significant.

**Strengths:**

1. The motivation is well justified, the method is reasonable.
2. The problems are practical and relevant to the large-scale MARL community.
3. Well-written paper with comprehensive results reported.
4. Authors provide several experimental details and corresponding code on their proposed algorithm.

**Weaknesses:**

While I lean towards acceptance, I have a few concerns. I am willing to raise my score if the authors can resolve my concerns and questions.
1. The novelty of Sable should be further clarified, stating how it builds upon, differs from, and improves upon existing methods, and so on.
2. As stated in the abstract, “In this work, we introduce Sable, a novel and theoretically sound algorithm that adapts the retention mechanism from Retentive Networks to MARL.” Where is the theoretical convergence guarantee analysis in the main text?

**Questions:**

I have asked my questions in the Weaknesses section. I have no further questions.

---

> ### Author Response · Authors · 2024-11-18
> **Author's reply to Reviewer 7eua**
>
> We thank the reviewer for their feedback and for highlighting the significance of Sable’s improvement on memory efficiency and agent scalability.
>
> > The novelty of Sable should be further clarified, stating how it builds upon, differs from, and improves upon existing methods, and so on.
>
> We thank the reviewer for raising this point and think our paper can be significantly improved by highlighting Sable’s novelty more explicitly than we currently do.
> The core contributions of Sable are as follows:
> * We extend the underlying RetNet architecture to be able to:
>   - Be encoder-decoder as the original RetNet work is decoder only.
>   - Enable RetNets to have cross-retention. The original work only has self-retention.
>   - We extend RetNets to have resettable hidden states over a temporal sequence to ensure that information does not flow across episode boundaries. This makes RetNets suitable for the reinforcement learning setting.
> * Moreover we improve upon the original MAT architecture by:
>   - Giving agents temporal memory over timesteps as MAT can only attend over agents one timestep at a time.
> * Further, Sable is the first algorithm to use RetNets to learn RL policies.
>
> Due to leveraging Retention, Sable also has the following advantages:
> * It has efficient memory usage over long sequences which allows it to scale to tasks with many agents. Something that MAT fails to do.
> * Due to the recurrent form of retention it has no need for caching sequences at act time. This means that it acts like an RNN at inference and maintains temporal memory through a hidden state instead of having to keep cached sequences like transformers do [txl2020].
>
> We will update the paper to reflect the novel contributions of Sable more explicitly and believe that it will benefit readers.
>
> > As stated in the abstract, “In this work, we introduce Sable, a novel and theoretically sound algorithm that adapts the retention mechanism from Retentive Networks to MARL.” Where is the theoretical convergence guarantee analysis in the main text?
>
> We kindly ask that the reviewer to refer to the global reply for a more detailed discussion on this point and for an overview on how we plan to adapt the main text for improved clarity.
>
> **References**
>
> [txl2020] Parisotto, Emilio, et al. "Stabilizing transformers for reinforcement learning." International conference on machine learning. PMLR, 2020.

---

> ### Author Response · Authors · 2024-11-25
> **Eager for engagement in discussion**
>
> Dear reviewer `7eua`, thank you again for your time and feedback. We have replied to your comments and revised our paper based on your feedback. We are eager to engage in discussion and we'd appreciate it if you could please reply to our response. We welcome any further comments and/or suggestions.

---

> > ### Comment · Reviewer_7eua · 2024-11-26
> >
> > Thank you for the authors' replies. If Sable's convergence guarantee is similar to HAPPO and MAT, then Sable's symbolic expression cannot reflect the important representation in the $Multi-Agent Advantage Decomposition$ Theorem. I think the authors should explain. Meanwhile, the page limitation is not satisfied. I will keep my rating.

---

> ### Author Response · Authors · 2024-11-26
>
> We thank the reviewer for engaging with us in discussion. We are not entirely sure what the reviewer means by *"Sable's symbolic expression cannot reflect the important representation in the Multi-Agent Advantage Decomposition Theorem"*. We gladly invite the reviewer to further elaborate on this point.
>
> If we had to interpret what we think the reviewer means by this, the reviewer might be trying to highlight a mismatch in the theoretical formulation of the theorem, which requires conditioning on a growing number of actions, and the formulation of the PPO objective in HAPPO/MAT as practical algorithms that use this theorem. To elaborate on this, we recapitulate from Kuba et al. (2021) that from Proposition 2 in their work, we can estimate $\mathbb{E}\_{a^{1:m} \sim \hat{\pi}^{1:m - 1}\_{\theta_{k + 1}}, a^m \sim \hat{\pi}^m\_{\theta_k}} \left[ A^{\pi_g}\_{\pi\_{\theta_k}} (s, a^{1:m - 1}, a^m) \right]$ with an estimator of the form
>
> $$
> \left( \frac{\hat{\pi}^m(a^m | s)}{\pi^m\_{\theta_k}(a^m | s)} - 1 \right) M^{1:m}(s,a), \text{ where } M^{1:m} = \frac{\hat{\pi}^{1:m - 1}(a^{1:m - 1} | s)}{\pi^{1:m - 1}(a^{1:m - 1} | s)} A(s,a).
> $$
> This means we can simply maintain a joint advantage function and use autoregressive policy updates. Furthermore, we assume that the joint policy factorises, which then for HAPPO we can write the update as
> $$
> \text{for agent } i_m = 1, \dots, n \text{ do} \\
> $$
> $$
> \quad \text{Update actor } \pi^m \text{ with } \theta^m_{k+1}, \text{ the argmax of the PPO-Clip objective} \\
> $$
> $$
> \quad \frac{1}{BT} \sum_{b=1}^B \sum_{t=0}^T \min \left( \frac{\pi^m_{\theta^m}(a^m_t | o^m_t)}{\pi^m_{\theta^m_k}(a^m_t | o^m_t)} M^{1:m}(s_t, a_t), \text{clip} \left( \frac{\pi^m_{\theta^m}(a^m_t | o^m_t)}{\pi^m_{\theta^m_k}(a^m_t | o^m_t)}, 1 - \epsilon, 1 + \epsilon \right) M^{1:m}(s_t, a_t)  \right). \\
> $$
> $$
> \quad \text{Compute } M^{1:m+1}(s,a) = \frac{\pi^m_{\theta^m_{k+1}}(a^m | o^m)}{\pi^m_{\theta^m_k}(a^m | o^m)} M^{1:m}(s,a).
> $$
> where $M^{i_1}(s, \mathbf{a}) = \hat{A}(s, \mathbf{a})$ and use the generalized advantage estimation (GAE) with
> $\hat{V_t} = \frac{1}{n} \sum_{m=1}^n V^i(o^i_t)$, as is also done in MAT. However, since MAT and Sable are by design autoregressive, the above update can be computed in parallel during training using masking.
>
> We want to reiterate that our optimisation objective and update scheme are identical to HAPPO in principle and to MAT in implementation, which both inherit theoretical guarantees from the mirror learning framework developed by Kuba et al. (2022). Furthermore, our architectural innovations have no bearing on the assumptions required within the mirror learning framework. Therefore if the reviewer is able to highlight a flaw in our construction, it should apply equally to this prior work.
>
> Finally, for the page limit, we quote the author guide for ICLR 2025: *"Authors are strongly encouraged to include a paragraph-long Reproducibility Statement at the end of the main text (before references)... This optional reproducibility statement will not count toward the page limit, but should not be more than 1 page."* Therefore, we hope the reviewer accepts that we are indeed still within the page limit.
>
> We look forward to further engagement and thank the reviewer again for their comments.
>
> **References**
>
> [Kuba2021] Kuba, Jakub Grudzien, et al. "Trust region policy optimisation in multi-agent reinforcement learning." arXiv preprint arXiv:2109.11251 (2021).
>
> [Kuba2022] Kuba, Jakub Grudzien, et al. "Heterogeneous-agent mirror learning: A continuum of solutions to cooperative marl." arXiv preprint arXiv:2208.01682 (2022).

---

### Official Review · Reviewer_BtD1 · 2024-11-01

**Soundness:** 4
**Presentation:** 4
**Contribution:** 4
**Rating:** 6
**Confidence:** 3

**Summary:**

This paper begins by analyzing the broad categorization of current MARL approaches and addresses the issues of high computational and memory resource costs and the lack of capability to capture historical information in MAT. Inspired by Retention, the paper adapts the Retention architecture for MARL, introducing the Sable method. This method maintains high computational efficiency and low memory consumption while supporting a large number of agents, and it leverages historical knowledge to improve policy learning. The paper concludes with a comparison of SOTA models in the MARL domain across multiple environments, achieving excellent results that thoroughly validate the algorithm's performance. Overall, this paper mitigates the problems inherent in traditional MAT methods and proposes a new, efficient MARL architecture, providing a fresh direction and approach for research in this field.

**Strengths:**

1. The introduction succinctly outlines the general classification of MARL methods and their limitations, while also identifying the issues with current SOTA methods. This leads smoothly into the presentation of the paper's method and its advantages, ensuring a coherent flow and quick understanding of the current state of MARL and the paper's contributions.
2. Building on the shortcomings of previous MAT methods, the paper innovatively proposes the Sable method by adapting the Retention architecture to RL algorithms. This adaptation results in a method with lower computational and memory costs, scalability to a large number of agents, and the ability to incorporate historical observations for better policy learning. The method also provides new research directions and theoretical foundations for the MARL domain.
3. The paper conducts experiments in several classic and challenging environments, comparing multiple SOTA models in the MARL domain. The extensive experimental results demonstrate the effectiveness of the algorithm, and additional experiments with varying numbers of agents confirm its computational and memory efficiency.

**Weaknesses:**

1. The Related Work section lacks a thorough introduction to some classic MARL algorithms, which would help readers unfamiliar with the field understand the broader development and methodologies in MARL.
2. While the Introduction directly addresses the classification of MARL methods and their current limitations, it could benefit from more foundational background information to better contextualize the motivation and overall method design for readers.

**Questions:**

1. Can the method adapt to changes in the number of agents or significant variations in observation while maintaining the robustness of the overall policy?

---

> ### Author Response · Authors · 2024-11-18
> **Author's reply to Reviewer BtD1**
>
> We thank the reviewer for their feedback and for taking the time to review our work. We would also like to thank the reviewer for giving our work the highest possible scores for soundness, presentation and contribution.
>
> >The Related Work section lacks a thorough introduction to some classic MARL algorithms, which would help readers unfamiliar with the field understand the broader development and methodologies in MARL.
>
> We will include classic MARL algorithms relevant to our problem setting in the related work to better contextualise Sable within the literature. This should result in the paper being more accessible to a wider audience and we thank the reviewer for this suggestion.
>
> >While the Introduction directly addresses the classification of MARL methods and their current limitations, it could benefit from more foundational background information to better contextualize the motivation and overall method design for readers.
>
> We thank the reviewer for this suggestion.  We will expand the related work and introduction of the paper to better position Sable and its novelty within the broader MARL landscape. We kindly refer the reviewer to our global reply for an overview of how we plan to include this in the main text for improved clarity.
>
> >Can the method adapt to changes in the number of agents or significant variations in observation while maintaining the robustness of the overall policy?
>
> We believe this should be possible. If care is taken to compute the per-agent advantage and to construct the decay matrices correctly, it should be possible to handle varying numbers of agents during execution. It should also be possible to handle varying agent observations as long as each agent’s observation is correctly projected into the embedding space of the underlying RetNet. While these are exciting and interesting research questions, we consider them as out of scope for our current work and will investigate these in future work.

---

> > ### Author Response · Authors · 2024-11-25
> > **Eager for engagement in discussion**
> >
> > Dear reviewer `BtD1`, thank you again for your time and feedback. We have replied to your comments and revised our paper based on your feedback. We are eager to engage in discussion and we'd appreciate it if you could please reply to our response. We welcome any further comments and/or suggestions.

---

### Official Review · Reviewer_3TrA · 2024-11-03

**Soundness:** 3
**Presentation:** 3
**Contribution:** 3
**Rating:** 6
**Confidence:** 4

**Summary:**

The paper introduces Sable, a novel algorithm for multi-agent reinforcement learning (MARL) that aims to achieve strong performance, memory efficiency, and scalability. Sable adapts the retention mechanism from Retentive Networks to MARL, allowing for computational efficiency and the ability to maintain long temporal contexts. The authors demonstrate that Sable outperforms several state-of-the-art methods across diverse environments.

**Strengths:**

1. This paper introduces a novel approach by adapting retention mechanisms for multi-agent reinforcement learning (MARL), representing a creative solution to enhance memory efficiency and scalability as the number of agents increases. The experimental results are particularly impressive.

2. The study includes a comprehensive set of experiments encompassing six commonly used MARL environments and a total of 45 tasks. The results demonstrate robust performance, indicating that Sable significantly outperforms existing methodologies in 34 out of 45 tasks. Furthermore, the approach exhibits stability with over a thousand agents, which is a critical consideration for large-scale applications.

**Weaknesses:**

The related work section is insufficiently comprehensive, primarily focusing on recurrent nets and Transformers. It would be beneficial to include more detailed discussions on specific algorithms in multi-agent reinforcement learning (MARL). In the MABrax environment, the performance of Sable is significantly worse compared to other environments. A more in-depth analysis and discussion of the reasons for this disparity would be highly valuable.

**Questions:**

1. Table 1 is too small, affecting the overall aesthetics and clarity of the presentation. Could the structure be adjusted or some content trimmed to enhance its visual appeal and readability?

2. The ablation study fails to pinpoint the main design elements that influence Sable's performance. Although the modified MAT shows some improvement, it still falls short compared to Sable. Further analysis and experimentation would likely be beneficial for the community to effectively reuse these insights.

3. In Figures 4b, 4c, and 4d, the final reward continues to increase as the number of agents increases. I would like to understand the reason behind this because usually the performance will decrease as the number of agents increasing.

---

> ### Author Response · Authors · 2024-11-18
> **Author's reply to Reviewer 3TrA - Part 1**
>
> We thank the reviewer for their feedback and will address their concerns and questions below.
>
> > The related work section is insufficiently comprehensive, primarily focusing on recurrent nets and Transformers. It would be beneficial to include more detailed discussions on specific algorithms in multi-agent reinforcement learning (MARL)
>
> We thank the reviewer for this suggestion which should improve the accessibility of the work. We will make these updates to the related work section and let the reviewer know once these updates have been made.
>
> > In the MABrax environment, the performance of Sable is significantly worse compared to other environments. A more in-depth analysis and discussion of the reasons for this disparity would be highly valuable.
>
> We agree that this is an important point to understand. Unfortunately, we are unable to offer much more insight than is already in the paper as it is an open direction of research we are currently pursuing. SAC has been shown to outperform PPO on Brax/MuJoCo [sac2018, brax2021] (the single agent version of this benchmark), this may be due to the off-policy nature of SAC, the entropy-based policy or other algorithmic differences. Given that PPO is the underlying algorithm behind both MAT and Sable, we believe this is a large contributor to the difference in performance. However, we want to highlight that Sable, on average, outperforms MASAC/HASAC on MPE, which is the other continuous-action space environment, as such it is unlikely to be the continuous nature of MaBrax, but rather the dynamics of the environment that favour the SAC-based algorithms.
>
> > Table 1 is too small, affecting the overall aesthetics and clarity of the presentation. Could the structure be adjusted or some content trimmed to enhance its visual appeal and readability?
>
> We will move the table from the main text to the appendix and replace it with a larger (in font size) table containing only the aggregated results over each environment suite. This should help make the table more readable.
>
> > The ablation study fails to pinpoint the main design elements that influence Sable's performance. Although the modified MAT shows some improvement, it still falls short compared to Sable. Further analysis and experimentation would likely be beneficial for the community to effectively reuse these insights.
>
> In the ablation study, we test the main differences between MAT and Sable. Using RMSNorm for normalisation and SwiGLU for the feedforward layers are the two **implementation** details that differ between the algorithms. The only remaining differences are (1) the transformer versus the RetNet and (2) the fact that Sable is able to condition on long sequences of past observations. Given the reviewer’s suggestion, we have decided to add Sable without temporal memory to our ablation. The updated plots can be found [here](https://sites.google.com/view/sable-marl/rebuttal-experiments) (Sable without memory is indicated with a dashed line). This shows that the improvements come from Sable’s temporal memory capabilities, rather than its use of retention, as Sable without memory performs comparably to MAT. We will include these updated plots in the paper and improve the discussion around this ablation to make it clear that the retention enables Sable’s performance, by allowing for long sequences of past observations, but it is not retention itself that is responsible for the performance improvements.
>
> > In Figures 4b, 4c, and 4d, the final reward continues to increase as the number of agents increases. I would like to understand the reason behind this because usually the performance will decrease as the number of agents increasing.
>
> We thank the reviewer for pointing this out. In Level-based foraging, the team is rewarded when agents collect food items. In Figures 4b, 4c and 4d as we increased  the number of agents we did not increase the grid size, leading to a greater density of agents in the world and thus a higher likelihood of agents finding food, resulting in a higher reward for the larger scenarios. This was not our intention, we wanted to keep the ratio of agents to grid size constant in these experiments, but by mistake we did not. We have rerun all these experiments while keeping a fixed agent to grid density. The relative ordering of algorithm performance has remained unchanged and the maximum reward now decreases as the number of agents increases. We believe this will more clearly illustrate the point to readers that Sable is able to effectively co-ordinate at scale while maintaining reasonable computational memory usage. We will update the paper with the results, in the interim, the updated results can be found at this [link](https://sites.google.com/view/sable-marl/rebuttal-experiments).

---

> ### Author Response · Authors · 2024-11-18
> **Author's reply to Reviewer 3TrA - Part 2**
>
> **References**
>
> [brax2021] Freeman, C. Daniel, et al. "Brax--a differentiable physics engine for large scale rigid body simulation." arXiv preprint arXiv:2106.13281 (2021).
> [sac2018] Haarnoja, Tuomas, et al. "Soft actor-critic algorithms and applications." arXiv preprint arXiv:1812.05905 (2018).

---

> > ### Author Response · Authors · 2024-11-25
> > **Eager for engagement in discussion**
> >
> > Dear reviewer `3TrA`, thank you again for your time and feedback. We have replied to your comments and revised our paper based on your feedback. We are eager to engage in discussion and we'd appreciate it if you could please reply to our response. We welcome any further comments and/or suggestions.

---

### Official Review · Reviewer_rdRb · 2024-11-03

**Soundness:** 2
**Presentation:** 2
**Contribution:** 2
**Rating:** 3
**Confidence:** 4

**Summary:**

The paper presents Sable, an approach based on a modified Retentive Network (RetNet) architecture that leverages attention mechanisms with a retention-based memory management approach. It is shown to improve the memory efficiency of MARL and scale well to large number of agents.

**Strengths:**

+ The proposed Stable considers MARL as a sequence modeling problem and leverages Retentive Networks. It is able to maintaining a long temporal context and leads to efficient memory usage.

**Weaknesses:**

While I like the idea presented in this paper, there are many issues with the experiments and evaluation.
- Formulating MARL as a sequence modeling problem has been an active area of research. The authors have cited (Wen et al., 2022) and  (Kuba et al., 2021). But since IPPO was considered as a main baseline, why not also consider baselines like independent decision transformers and other sequence modeling approaches? Further, there are also many decomposition based MARL algorithms that outperform QMIX (e.g., weighted-QMIX, FOP, DOP, PAC) and mean-field based MARL algorithms. These should not be ignored in the evaluation.
- In Figure 1 - Middle, while the authors claim that Stable is more throughput efficient, but it is only 6.5x, compared to 45x by IPPO.
- In Figure 1 - Right, IPPO is 9x but the bar is much lower than Stable that is 7x.
- In Figure 4, IPPO seems to consistently have lower memory usage than Stable. Doesn't it conflict with the main claim of the paper?
- In Table 1,  MASAC, HASAC, and QMIX have many missing return values. It is hard to believe these algorithms don't work in these environments.
- The comparison with baseline algorithms need to cover more complex MARL tasks with continuous action/state spaces, like Starcraft and MuJoCo, which are used by many of the baselines.
- Table 1 summarizes the return comparison with baselines. Since the main advantage of Stable is memory efficiency, a similar table on memory comparison should be provided.

Further, the authors claim that Sable provides "theoretically grounded convergence guarantees". But I was not able to find any analysis or proofs in the paper. It simply cited (Kuba et al., 2021). It is unclear how it applies here.

**Questions:**

Please address my questions in the weakness section.

---

> ### Author Response · Authors · 2024-11-18
> **Author's reply to Reviewer rdRb - Part 1**
>
> We thank the reviewer for their time reviewing our work. We directly address the reviewer’s concerns below:
>
> >Formulating MARL as a sequence modeling problem has been an active area of research. The authors have cited (Wen et al., 2022) and (Kuba et al., 2021). But since IPPO was considered as a main baseline, why not also consider baselines like independent decision transformers and other sequence modeling approaches?
>
> Indeed, this is an active area of research. We wish to highlight to the reviewer that the decision transformer [dt2021], and by extension the multi-agent decision transformer [madt2021], are offline reinforcement learning algorithms. In our work, we focus on the online MARL case. This makes the approach infeasible for our comparisons.
>
> > Further, there are also many decomposition based MARL algorithms that outperform QMIX (e.g., weighted-QMIX, FOP, DOP, PAC) and mean-field based MARL algorithms
>
> It has been shown that a well-tuned QMIX outperforms various extensions to QMIX [riit2021] for this reason we feel that QMIX represents a sufficient off-policy baseline.
>
> > Clarifying questions around Figures 1 and 4
>
> We wish to clarify and remove confusion regarding the interpretation of both Figure 1 and 4. Our research objective is to develop an algorithm with state-of-the-art performance that is at the same time significantly more memory efficient than the current best-performing algorithms. Specifically, IPPO is efficient but not the best performing, MAT has good performance but is not efficient, Sable provides the best performance of the three, at a memory efficiency that is far superior to MAT but not as good as IPPO. We comment on each specific figure below:
>
> In Figure 1, our aim is to highlight that Sable achieves the best performance across most tasks while maintaining good steps per second (SPS) and memory usage. Our primary comparison was with MAT, as it is the most performant algorithm in the literature. To ensure a fair and comprehensive comparison, we included the performance bar for all other baselines (excluding MAT and Sable) in the left bar plot and specifically added IPPO to the SPS and memory comparisons, as it is the fastest and most memory-efficient baseline. This inclusion is intended to transparently illustrate that while Sable does not achieve the highest SPS among existing algorithms, it effectively balances performance, memory efficiency, and speed. In Figure 4, while IPPO can scale to many agents, it is unable to learn effectively in such scenarios, as proven by the instability in the mean episode return plots within this figure. We direct the reviewer to the results paragraph in Section 4.2 (Lines 463-471) for a more detailed discussion.
>
> >In Table 1, MASAC, HASAC, and QMIX have many missing return values. It is hard to believe these algorithms don't work in these environments.
>
> SAC is a continuous action space algorithm and does not work on discrete action spaces [sac2018] without significant adaptation [disc-sac2019, disc-sac2022], these adaptations have not been thoroughly tested in MARL, which is why we opt to not use these as baselines. By extension, MASAC and HASAC inherent this property which is why we only evaluate these algorithms on the continuous action space tasks. Further, we omit QMIX from other discrete action space tasks since it has been shown to perform very poorly on Robotic Warehouse and Level-based foraging [uoe-bench2020]. We keep QMIX as a SMAX baseline due to its strong historical performance on the benchmark [riit2021].
>
> > The comparison with baseline algorithms need to cover more complex MARL tasks with continuous action/state spaces, like Starcraft and MuJoCo, which are used by many of the baselines.
>
> We have already evaluated all algorithms on MaBrax and SMAX [jaxmarl] which are the JAX-based implementations of MaMuJoCo and Starcraft Mulit-Agent Challenge (SMAC) respectively. Please see the results in Table 1 and Figures 3b and 3c. We will make this point more clear in the paper and not only in the appendix.
>
> > Table 1 summarises the return comparison with baselines. Since the main advantage of Stable is memory efficiency, a similar table on memory comparison should be provided.
>
> In Figure 4 we illustrate that when there are few agents, the difference in memory requirements between algorithms is rather small. For this reason, we only show Sable’s memory efficiency over MAT on larger agent scales and believe adding a table of memory usage will not be particularly beneficial to the reader.

---

> ### Author Response · Authors · 2024-11-18
> **Author's reply to Reviewer rdRb - Part 2**
>
> > Further, the authors claim that Sable provides "theoretically grounded convergence guarantees". But I was not able to find any analysis or proofs in the paper. It simply cited (Kuba et al., 2021). It is unclear how it applies here.
>
> We thank the reviewer for highlighting confusion around this point. We kindly refer the reviewer to our global reply for further clarification and discussion. We will make sure to improve the clarity on this aspect of the work in the paper.
>
> **References**
>
> [riit2021] Hu, Jian, et al. "Rethinking the implementation tricks and monotonicity constraint in cooperative multi-agent reinforcement learning." arXiv preprint arXiv:2102.03479 (2021).
> [mat2022] Wen, Muning, et al. "Multi-agent reinforcement learning is a sequence modeling problem." Advances in Neural Information Processing Systems 35 (2022): 16509-16521.
> [uoe-bench2020] Papoudakis, Georgios, et al. "Benchmarking multi-agent deep reinforcement learning algorithms in cooperative tasks." arXiv preprint arXiv:2006.07869 (2020).
> [dt2021] Chen, Lili, et al. "Decision transformer: Reinforcement learning via sequence modeling." Advances in neural information processing systems 34 (2021): 15084-15097.
> [madt2021] Meng, Linghui, et al. "Offline pre-trained multi-agent decision transformer: One big sequence model tackles all smac tasks." arXiv preprint arXiv:2112.02845 (2021).
> [jaxmarl] Rutherford, Alexander, et al. "Jaxmarl: Multi-agent rl environments in jax." arXiv preprint arXiv:2311.10090 (2023).
> [kuba2021] Kuba, Jakub Grudzien, et al. "Settling the variance of multi-agent policy gradients." Advances in Neural Information Processing Systems 34 (2021): 13458-13470.
> [sac2018] Haarnoja, Tuomas, et al. "Soft actor-critic algorithms and applications." arXiv preprint arXiv:1812.05905 (2018).
> [disc-sac2019] Christodoulou, Petros. "Soft actor-critic for discrete action settings." arXiv preprint arXiv:1910.07207 (2019).
> [disc-sac2022] Zhou, Haibin, et al. "Revisiting discrete soft actor-critic." arXiv preprint arXiv:2209.10081 (2022).

---

> ### Author Response · Authors · 2024-11-25
> **Eager for engagement in discussion**
>
> Dear reviewer `rdRb`, thank you again for your time and feedback. We have replied to your comments and revised our paper based on your feedback. We are eager to engage in discussion and we'd appreciate it if you could please reply to our response. We welcome any further comments and/or suggestions.

---

### Author Response · Authors · 2024-11-18
**Author's reply to all Reviewers - Part 1**

**We thank all the reviewers for their time and for reviewing our work.** We will respond to each reviewer’s questions individually to address their concerns and give suggestions for changes based on their comments. Once any updates have been made, we will specifically highlight these sections in revised versions of the paper. For overall comprehension and to make the discussion easy to initiate, **we wish to summarise positives** on what reviewers had to say about the work and **point out shared concerns that we aim to address**:

## Shared praise

Reviewer `3TrA` writes that the introduced algorithm represents

>… a creative solution to enhance memory efficiency and scalability as the number of agents increases.

that

> The experimental results are particularly impressive.

and finally that the paper’s results

>… demonstrate robust performance, indicating that Sable significantly outperforms existing methodologies in 34 out of 45 tasks. Furthermore, the approach exhibits stability with over a thousand agents, which is a critical consideration for large-scale applications.

Reviewer `BtD1` gives the work the highest possible scores for Soundness, Presentation and Contribution (4 for all) and says that the paper is written in a way that ensures a

> …coherent flow and quick understanding of the current state of MARL and the paper's contributions

that

> …the paper innovatively proposes the Sable method by adapting the Retention architecture to RL algorithms. This adaptation results in a method with lower computational and memory costs, scalability to a large number of agents, and the ability to incorporate historical observations for better policy learning.

and that

> … extensive experimental results demonstrate the effectiveness of the algorithm, and additional experiments with varying numbers of agents confirm its computational and memory efficiency.

Reviewer `7eua` writes that

> … the advancements of Sable in memory efficiency and agent scalability are significant

that the problems addressed in the paper

> …are practical and relevant to the large-scale MARL community.

and that the work is a

> Well-written paper with comprehensive results reported

## Shared concerns

As for concerns, we wish to highlight the following points that were raised by more than one reviewer.

**1. Missing theoretical convergence guarantees analysis**

Reviewer `rdRb` writes that

>…the authors claim that Sable provides "theoretically grounded convergence guarantees". But I was not able to find any analysis or proofs in the paper. It simply cited (Kuba et al., 2021). It is unclear how it applies here.

Reviewer `7eua` asks

>…Where is the theoretical convergence guarantee analysis in the main text?

We thank the reviewers for raising this issue, as admittedly it was not clearly outlined within the main text.

------------------------------------------------------------------------------------------------------------------------------------------------

**Updated in revised paper**: we have added a detailed clarification of this point in the main text of the revised version of our paper which has been uploaded. We hope this will address the reviewer's concerns and we remain open to any additional feedback that could improve our manuscript.

------------------------------------------------------------------------------------------------------------------------------------------------

*(Older reply): Our claim relies on previous theoretical results proved in a series of recent papers. First, is the advantage decomposition theorem/lemma [kuba2021, Lemma 1], as highlighted in our paper. This result was subsequently used to develop trust region learning approaches for MARL with monotonic performance improvement guarantees [kuba2022a, Theorem 2]. Even though PPO-style algorithms with autoregressive updates do not strictly adhere to trust region learning theory, and therefore, do not have strict monotonic improvement, more recent work has placed PPO within a class of mirror learning algorithms that indeed enjoy monotonic improvement and theoretical convergence guarantees [kuba2022b, Theorem 3.6]. This work has since been extended to the multi-agent setting [kuba2022c, Theorem 1], and in particular, HAPPO (multi-agent PPO with heterogeneous autoregressive updates), was shown to be an instance of multi-agent mirror learning, and therefore theoretically sound. To obtain an instance of mirror learning requires defining a valid drift functional, neighbourhood operator and sampling distribution. In both MAT and Sable, these design choices are exactly as they are for HAPPO, and therefore, we claim that Sable inherits the same theoretical monotonic improvement and convergence guarantees as HAPPO, and by extension MAT.*

We will make sure to update our paper to significantly improve the text around these details, as well as recapitulate certain core theoretical results and derivations supporting our claim in the appendix.

---

> ### Author Response · Authors · 2024-11-18
> **Author's reply to all Reviewers - Part 2**
>
> **2. Insufficiently detailed related work**
>
> Reviewer `3TrA` writes that
>
> > …The related work section is insufficiently comprehensive, primarily focusing on recurrent nets and Transformers. It would be beneficial to include more detailed discussions on specific algorithms in multi-agent reinforcement learning (MARL).
>
> Reviewer `BtD1` writes that
>
> > …The Related Work section lacks a thorough introduction to some classic MARL algorithms, which would help readers unfamiliar with the field understand the broader development and methodologies in MARL.
>
> and
>
> > …While the Introduction directly addresses the classification of MARL methods and their current limitations, it could benefit from more foundational background information to better contextualize the motivation and overall method design for readers.
>
> Reviewer `7eua` writes that
>
> >…The novelty of Sable should be further clarified, stating how it builds upon, differs from, and improves upon existing methods, and so on.
>
> We thank the reviewers for highlighting this shortcoming of our presentation, we will expand the introduction and the related work section to better position Sable among current MARL methods, including seminal work as well as recent publications.
>
> ------------------------------------------------------------------------------------------------------------------------------------------------
>
> **Updated in revised paper**: we have significantly expanded the introduction with additional context, related work and more clearly explaining Sable's position and novelty within the existing literature. We hope this will address the reviewer's concerns and we remain open to any additional feedback that could improve our manuscript.
>
> ------------------------------------------------------------------------------------------------------------------------------------------------
>
> *(We removed the older related work sketch provided in an earlier reply for improved readability)* ~To this end, we briefly provide here a sketch of the MARL landscape and Sable’s position within it, which we will expand on in the paper (note we cite some core papers below, which are not exhaustive. We will expand on this list and...~
>
> We welcome any additional suggested works that are particularly relevant from reviewers.

---

### Author Response · Authors · 2024-11-22
**New revised version of our paper addressing main concerns**

Dear reviewers, we want to thank you all once again for your time reviewing our work and for providing constructive feedback to improve our manuscript. **Based on your feedback and the main concerns raised, we have revised and uploaded a new version of our paper. We made the following changes**:

* We have significantly expanded the introduction with additional context, related work and more clearly explaining Sable's position and novelty within the existing literature. *[Addressing concerns raised by `rdRb` and `7eua`]*
* We have added detailed clarification in the main text of how Sable has theoretical monotonic improvement and convergence guarantees. *[Addressing concerns raised by `3TrA `, `BtD1 ` and `7eua`]. Note: We will still add additional theoretical derivations (recapitulating existing results) in the appendix to further clarify this point.*

We also added the following to address specific concerns raised by `3TrA `:
* We expanded on and added references supporting why Sable performs worse than SAC-based algorithms on MaBrax.
* We added "Sable without memory" to the ablation study and significantly improved the discussion around the reason for Sable's impressive performance. We now fully isolate that the reason for Sable's performance is the way it structures the sequence enabling its long-term memory.
* We updated Table 1 to only have per-environment aggregations for all algorithms to make the table more readable and to improve the paper's overall aesthetics. We include the per-task tables in the Appendix and refer the reader to them in the main text for more results.
* We updated Figures 4b, 4c, and 4d to have constant agent-to-gird ratios for all tasks leading to the maximum reward per task decreasing as the number of agents increases.

We hope to have addressed all the reviewer's main concerns and that reviewers find our revised paper a satisfactory improvement. We remain open to any additional feedback that could improve our manuscript.

---

### Meta-Review · Area_Chair_MJj2 · 2024-12-19

**Metareview:**

This paper proposes Sable, a MARL algorithm that adapts retention mechanisms from Retentive Networks aiming to achieve memory efficiency and scalability. While the reviewers acknowledged the potential value of improving MARL performance with limited memory usage, significant concerns were raised about both theoretical and empirical aspects of the work. The paper claims convergence guarantees but lacks rigorous proofs, merely citing previous results without establishing how they extend to Sable's specific setting. The novelty needs better positioning as sequence modeling approaches and RetNet adaptations have been explored in MARL before. The experimental results show concerning inconsistencies - the claimed memory efficiency advantages appear contradicted by figures showing IPPO with lower memory usage, and performance on MaBrax environments notably lags behind baselines. Although the authors provided detailed responses and additional experiments, these core issues around theoretical soundness, technical novelty, and empirical validation remain insufficiently addressed.

Given these substantial limitations in the current manuscript, I recommend rejection.

**Additional Comments On Reviewer Discussion:**

While the reviewers acknowledged the potential value of improving MARL performance with limited memory usage, significant concerns were raised about both theoretical and empirical aspects of the work. The paper claims convergence guarantees but lacks rigorous proofs, merely citing previous results without establishing how they extend to Sable's specific setting. The novelty needs better positioning as sequence modeling approaches and RetNet adaptations have been explored in MARL before. The experimental results show concerning inconsistencies - the claimed memory efficiency advantages appear contradicted by figures showing IPPO with lower memory usage, and performance on MaBrax environments notably lags behind baselines. Although the authors provided detailed responses and additional experiments, these core issues around theoretical soundness, technical novelty, and empirical validation remain insufficiently addressed.

---

### Decision · Program_Chairs · 2025-01-22

Reject